# LETI: LEARNING TO GENERATE FROM TEXTUAL INTERACTIONS

## ABSTRACT

Finetuning pre-trained language models (LMs) is essential for enhancing their capabilities and is a crucial phase in their lifecycles. Existing techniques commonly fine-tune on input-output pairs (e.g., instruction fine-tuning Wei et al., 2022a) or with numerical rewards that gauge the output quality (e.g., reinforcement learning from human feedback Ouyang et al., 2022). We explore LMs' potential to **le**arn from **t**extual **i**nteractions (**LETI**) that not only check their correctness with binary labels but also pinpoint and explain errors in their outputs through textual feedback. Our focus is the code generation task, where the model produces code based on natural language instructions. This setting invites a natural and scalable way to acquire textual feedback: the error messages and stack traces from code execution using a Python interpreter. LETI iteratively fine-tunes the model, using the LM objective, on a concatenation of natural language instructions, LM-generated programs, and textual feedback, which is only provided when the generated program fails to solve the task. Prepended to this fine-tuning text, a binary reward token is used to differentiate correct and buggy solutions. LETI requires *no* ground-truth outputs for training and even outperforms a fine-tuned baseline that does. LETI not only improves the performance of two base LMs of different scales on a code generation dataset MBPP, but also generalizes to other datasets. Trained on MBPP, it achieves comparable or better performance than the base LMs on unseen problems in HumanEval. Furthermore, compared to binary feedback, we observe that textual feedback leads to improved generation quality and sample efficiency, achieving the same performance with fewer than half of the gradient steps. LETI is equally applicable in natural language tasks when they can be formulated as code generation, which we empirically verified on event argument extraction.[1]

## 1 INTRODUCTION

Large-scale language models have fundamentally shifted the paradigms of natural language processing (NLP). Based on LMs pre-trained on raw text, subsequent fine-tuning stages have proven crucial to enhance their capabilities in solving benchmark NLP tasks and generating texts that align with human preferences. Success has been achieved by fine-tuning with direct training signals that measure whether the model, e.g., classifies the input into the right category Devlin et al. (2019), answers a question correctly Li et al. (2017); Ramamurthy et al. (2022), summarizes documents well Stiennon et al. (2020); Wu et al. (2021), and generates outputs that align with human preferences Ouyang et al. (2022); Korbak et al. (2023). We hypothesize that LMs can harness the much richer training signals from textual interactions with the environment (e.g., a human or a Python interpreter) that not only *check the correctness* of LM's outputs but also *pinpoint the errors and explain why*.

We propose LETI, a new LM fine-tuning paradigm that aims to explore LMs' potential to **le**arn from nuanced **t**extual **i**nteractions. We evaluate LETI on code generation tasks, where the LM is supposed to generate code pieces to solve tasks described in natural language. This setting invites a natural and scalable way to acquire *automatic* interactive textual feedback: the stack traces and error message outputs by established programming language (PL) tools such as a Python interpreter. LETI's improvement process naturally mirrors a typical software development cycle: a human developer writes an initial program, executes it, and improves the program based on feedback obtained from

---

[1]Our code will be available at `<anonymized>`.

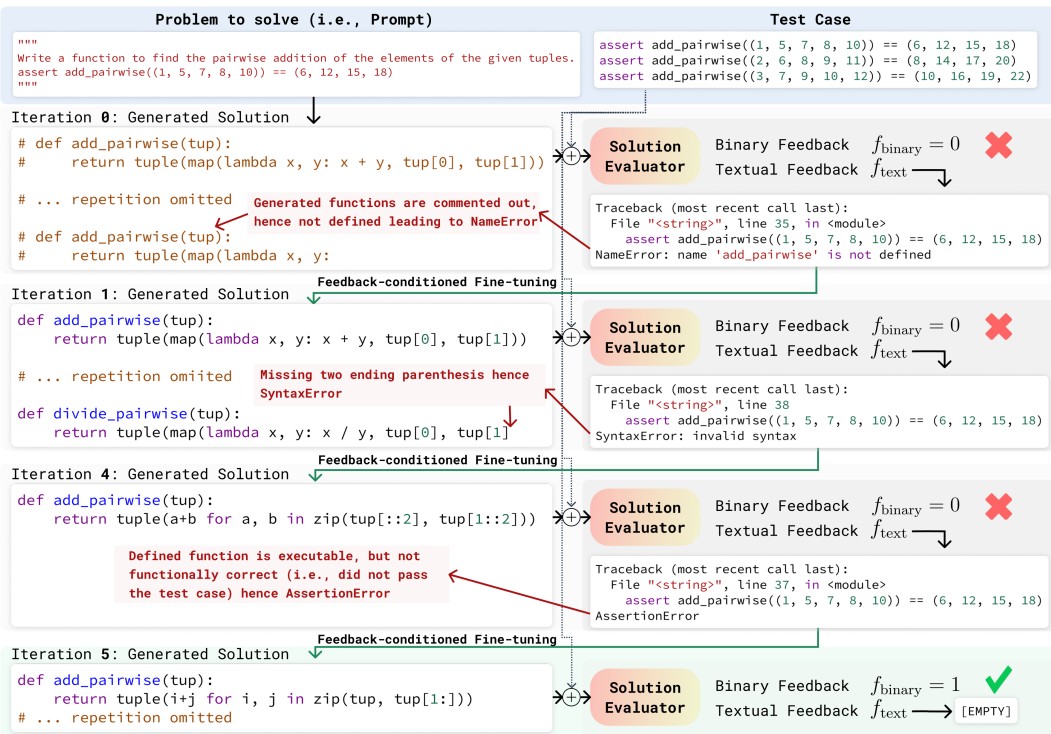

Figure 1: Qualitative example of LETI improving an LM on code generation by leveraging feedback from a solution evaluator (e.g., a Python interpreter). At each LETI iteration, the LM is first asked to generate candidate solutions. As a case study, we obtain binary and textual feedback by executing the solution against test cases using a Python interpreter. Feedback and the generated solutions are used to improve the LM generator for the next LETI iteration through feedback-conditioned fine-tuning (§2.3). This is a code generation (MBPP; Austin et al., 2021) test set example generated by a 2B model optimized with LETI. We omit a few iterations and repetitive code for clarity.

the programming environment until a satisfying solution is found (e.g., successfully executed with no error); Furthermore, the human developer learns from mistakes in this process and becomes a (slightly) better developer who can avoid similar mistakes in the future. Similarly to the human development process, we provide empirical evidence that LETI can learn from past mistakes and avoid similar errors in §3.2.

In LETI, a base LM pre-trained on both natural language and code[2] is asked to generate a piece of program conditioning on the natural language instruction, which is then tested on a suite of test cases. LETI fine-tunes the model on a concatenation of natural language instruction, LM-generated program, and the textual feedback (e.g., stack traces and error messages) that pinpoints the bug, which is only provided when the generated program fails to solve the task. In addition to textual feedback, we prepend the fine-tuning sequences with a reward token (i.e., binary feedback), which differs for correct (`<|good|>`) and buggy solutions (`<|bad|>`), to encourage the LM to generate correct solutions when conditioning on `<|good|>`. LETI repeats this procedure for multiple rounds. During this iterative process, LETI assumes *no* instruction-code paired data.

We find that LETI improves LM's performance on code generation tasks in MBPP (Austin et al., 2021) *without* using any ground-truth code. Specifically, it generates $63.2\%$ more syntactically correct and executable code (on the 2B LM) compared to the pre-trained model without any commonly employed post-processing heuristics[3]. When post-processing is applied, LETI (2B) improves performance and eliminates most NameError issues that occur when a variable or function is not defined (from 10% to 1%, on the 2B LM) in two iterations. The optimized LM also shows generalized performance

---

[2]Almost all modern large language models train on both natural language and code (Brown et al., 2020; OpenAI, 2023; Chowdhery et al., 2022; Touvron et al., 2023a).

[3]Stop-word-based post-processing heuristics (Fig. A.11) are commonly used by Code-LM (Chen et al., 2021b) to remove irrelevant code (e.g., only keep the first block of generated code).

improvement on another code generation dataset HumanEval (Chen et al., 2021b) (§3.2). Such improvement in in-domain tasks does *not* come at the cost of the capability of the original LM (e.g., reasoning and chain-of-thought capability Wei et al. 2022b) due to LETI's auxiliary objective that continuing pre-train along with fine-tuning (§3.4).

We observe that textual feedback is advantageous in terms of improving the LM compared to baselines that only use binary feedback, as it offers enhanced performance and greater sample efficiency that only requires about half of the gradient steps to reach the same performance for the 2B-scale model (§3.3). Furthermore, we find LETI is equally applicable to NLP tasks (e.g., event argument extraction Wang et al. 2023a) when they can be formulated into a code generation problem (§3.5).

## 2   LETI: LEARNING FROM TEXTUAL INTERACTIONS

Each iteration, LETI prompts the LM (§2.1) with the natural language problem description to generate a set of $n$ solutions. The solutions are then evaluated on a suite of test cases by a **Solution Evaluator** (§2.2) to generate textual feedback (i.e., stack traces and error messages). This work uses a Python interpreter as the solution evaluator to assess LM-generated solutions. The textual feedback is used to fine-tune the LM with **Feedback-Conditioned Fine-Tuning** (FCFT, §2.3).

We assume no ground-truth solutions while fine-tuning the LM, as LETI directly learns from solution evaluator's feedback. Intuitively, FCFT leverages textual feedback to associate various types of errors (e.g., `SyntaxError`) and solutions that commit them. Furthermore, with binary feedback, FCFT aligns correct or wrong solutions with corresponding pre-pended reward tokens `<|good|>` or `<|bad|>`, so that better solutions can be sampled from a trained LM by conditioning it on `<|good|>`. The workflow (one iteration) is described in Algorithm 1 and Fig. A.6.

### 2.1   LANGUAGE MODEL

The base LM can be any generative language model $p_\theta$, pre-trained on both natural and programming languages. For a given problem $x_i \in \mathcal{P}$, we sample $n$ solutions $\mathcal{S}_i = \{\hat{y}_{i,1}, \ldots, \hat{y}_{i,n}\}$ from $p_\theta(\cdot \mid x_i)$ (conditioned on reward token `<|good|>` when $p_\theta$ is fine-tuned for at least one iteration using FCFT), where each solution $\hat{y}_{i,j}$ is a sequence of tokens. We analyze the importance of problem set size $|\mathcal{P}|$ and the number of sampled solutions $n$ in §B.2 and §B.1. Since $p_\theta$ is trained on code, we assume that it can generate programs reasonably well in the training problem set, and at least some of the $n$ solutions are correct when an arbitrarily large $n$ is chosen. We use $n = 128$ for code generation experiments on MBPP (§3.2) and $n = 64$ for event argument extraction (§3.5).

### 2.2   SOLUTION EVALUATOR

Given a problem $x_i$, its test cases $\mathcal{T}_i$, and any generated solution $\hat{y}_{i,j}$, the Solution Evaluator $\phi$ (a Python interpreter) provides feedback $F_{i,j}$, which consists of binary $f_{\text{binary}}$ and textual feedback $f_{\text{text}}$ (i.e., $f_{\text{binary}}, f_{\text{text}} = \phi(x_i, \hat{y}_{i,j}, \mathcal{T}_i)$). $f_{\text{binary}} \in \{0, 1\}$ reflects the correctness of a solution, where $f_{\text{binary}} = 1$ means the given solution $\hat{y}_{i,j}$ can successfully solve the given problem $x_i$, and vice versa. $f_{\text{text}}$ is a concatenation of stack traces and a textual error message provided by the Python interpreter only when the generated solution commits an error on a test case. Examples of $f_{\text{text}}$ can be found in Fig. 1 and A.6. Generally speaking, we can implement $\phi$ differently for different types of problems; In §3.5, we show that it is possible to implement a $\phi$ that works for an NLP task.

### 2.3   FEEDBACK-CONDITIONED FINE-TUNING (FCFT)

Each LETI iteration samples solutions from LM $p_\theta$, evaluates generated solutions to obtain feedback using $\phi$, and improves the generator LM with feedback-conditioned fine-tuning (FCFT). FCFT fine-tunes $p_\theta$ on each problem $x_i$ and generated solution $\hat{y}_{i,j}$ conditioned on feedback $F_{i,j}$ (a sequence of tokens comprised of binary $f_{\text{binary}}$ and textual feedback $f_{\text{text}}$). This resembles on-policy reinforcement learning, where $p_\theta$ is the policy and the solution evaluator $\phi$ plays the role of a reward function.

Feedback $F_{i,j}$ concatenates one initial reward token that denotes the binary feedback $f_{\text{binary}}$ indicating whether the solution is correct, and textual feedback $f_{\text{text}}$, if provided. If the solution evaluator $\phi$ finds solution $\hat{y}_{i,j}$ correct, we use a reward token `<|good|>`, and `<|bad|>` otherwise. Follow-

ing the initial reward token, we include the textual feedback $f_{\text{text}}$, if provided, enclosed by two special tokens denoting the beginning and end of textual feedback (i.e., `<|text_feedback|>`, `<|/text_feedback|>`). That is, both feedback for the problem $x_i$ and solution $\hat{y}_{i,j}$ are a concatenated sequence of tokens: $F_{i,j} = f_{\text{binary}} \oplus \texttt{<|text\_feedback|>} \oplus f_{\text{text}} \oplus \texttt{<|/text\_feedback|>}$. In the case when $f_{\text{text}}$ is not provided (e.g., when $f_{\text{binary}} = 1$), only the initial reward token is included as feedback: $F_{i,j} = f_{\text{binary}}$. We expand the vocabulary of the initial pre-trained LM $p_\theta$ to include these additional tokens.

LETI optimizes $p_\theta$ with the language modeling objective on sequence $s = F_{i,j} \oplus x_i \oplus \hat{y}_{i,j}$ (i.e., a concatenation of instruction and generated solution conditioned on the feedback) as shown in part (1) of equation 1. A concrete example of a data instance can be found in Fig. A.6.

## 2.4 REGULARIZATION WITH CONTINUED PRE-TRAINING

To alleviate distribution shifts that may be caused by fine-tuning on generated solutions, we interleave FCFT optimization (§2.3) with LM objective optimization on the pre-training data. equation 1 puts the entire LETI's training loss together. Our ablation study shows that the regularization by continued pre-training is essential to maintain LM's original capability on tasks that it was not trained on (§3.4).

$$\mathcal{L}_{\text{LM}}(x, \theta) = -\sum_t \log p_\theta(x_t \mid x_{<t})$$

$$\mathcal{L}(\theta) = \underbrace{\frac{1}{|D_{\text{FCFT}}|} \sum_{s = F \oplus x \oplus \hat{y} \in D_{\text{FCFT}}} \mathcal{L}_{\text{LM}}(s, \theta)}_{\text{(1) Feedback-conditioned Fine-tuning (FCFT)}} + \underbrace{\frac{1}{|D_{\text{pre-train}}|} \sum_{s' \in D_{\text{pre-train}}} \mathcal{L}_{\text{LM}}(s', \theta)}_{\text{(2) Regularization with pre-training dataset}} \quad (1)$$

---

**Algorithm 1** One iteration of LETI Improvement using Feedback-conditioned Fine-tuning (FCFT).

---
**Require:** $D_{\text{pre-train}}$                                           ▷ Pre-training Dataset
    $D_{\text{FCFT}} \leftarrow \{\}$                                            ▷ Dataset for FCFT
    **for** each problem $x_i \in P$ and its test cases $\mathcal{T}_i$ **do**
        **for** $j = 1$ to $n$ **do**
            Sample a solution $\hat{y}_{i,j}$ from $p_\theta(\cdot \mid x_i)$, conditioned on `<|good|>` for fine-tuned $p_\theta$ (§2.1)
            $f_{\text{binary}}, f_{\text{text}} \leftarrow \phi(x_i, \hat{y}_{i,j}, \mathcal{T}_i)$          ▷ Generate feedback using evaluator $\phi$ (§2.2)
            $F_{i,j} = f_{\text{binary}} \oplus \texttt{<|text\_feedback|>} \oplus f_{\text{text}} \oplus \texttt{<|/text\_feedback|>}$
            $D_{\text{FCFT}} \leftarrow D_{\text{FCFT}} \cup \{F_{i,j} \oplus x_i \oplus \hat{y}_{i,j}\}$    ▷ Construct the feedback-conditioned dataset
        **end for**
    **end for**
    Fine-tune the LM $p_\theta$ for a fixed epochs on $D_{\text{FCFT}}$ and $D_{\text{pre-train}}$ (equation 1)

---

# 3 EXPERIMENTAL RESULTS

## 3.1 EXPERIMENT SETUP

**Base model.** We experiment with `CodeGen-mono` LMs (Nijkamp et al., 2022), a series of open-sourced LMs pre-trained with both natural language and code with a range of model sizes. The NL and PL mixture of pre-training data makes it possible to evaluate LETI on both NL and PL tasks. Due to limited computational resources, we choose to experiment with `350M` and `2B` sized models.

**Dataset for continued pre-training.** We use the `Python` subset of `TheStack v1.1` dataset (Kocetkov et al., 2022) as the continued pre-training dataset for the mixture pre-train objective (§2.4)[4].

## 3.2 LETI MAKES LMs BETTER CODE GENERATORS

### 3.2.1 MOSTLY BASIC PYTHON PROBLEMS (MBPP)

**Setup.** We use the **Mostly Basic Python Problems (MBPP)** dataset (Austin et al., 2021) for training and evaluation. It contains 974 short Python problems described in natural language targeting entry-level programmers. LETI requires *no* ground-truth code but assumes a test suite for each problem

---

[4]The pre-training dataset BIGPYTHON of `CodeGen-mono` is not publicly available at the time of writing.

that MBPP provides to check solutions' correctness. Additional details (e.g., hyper-parameters) can be found in §C. We allow the model to generate 512 tokens at max for each problem and evaluate the generated solutions by executing them against a test suite.

**Post-Processing.** Stop-word-based post-processing heuristics (Fig. A.11) are commonly employed by Code-LM (Chen et al., 2021b) to remove irrelevant code (e.g., only keep the first block of generated code) and improve performance. However, such post-processing heuristics require manual effort and are less scalable to extend to different tasks. Whether or not LMs can improve code generation without postprocessing is a great testbed to evaluate their capabilities of learning from textual feedback and is central to answering our research question. Therefore, we test the general applicability of LETI both with and without postprocessing. Unless otherwise noted, we default to without post-processing setting in the following experiments.

**Evaluation metrics.** We use the `pass@k` metric. The model generates $k$ solutions for each problem; it is considered successfully solving the problem if *at least* one of the $k$ solutions passes all test cases. With higher $k$ values, the chance of observing a correct output for a problem increases. To reduce variances, we sample more than $k$ solutions to estimate `pass@k`, see §C.1 for details.

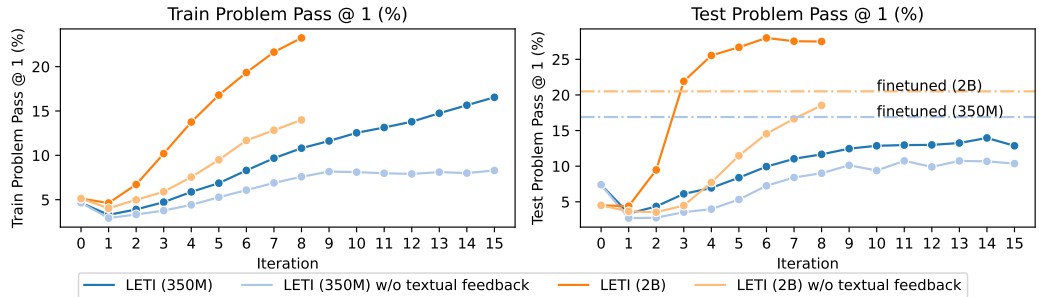

Figure 2: LETI (w/o post-processing) improves the base LMs performance on a code generation dataset MBPP. (left) LETI can iteratively improve the success rate of the LMs' generated solutions on training set problems; (right) LETI reaches performance close to (350M) or surpasses (2B) fine-tuned performance on the test set after a few iterations, despite not using any ground truth solutions.

**Results.** As shown in Fig. 2, LETI (w/o post-processing) learns from interactions with MBPP training set problems (i.e., iteratively generate, evaluate solutions, and learn from textual feedback) to generate better solutions for both *training* and *testing* problems. Despite not being fine-tuned on any ground truth solutions, LETI improves test set Pass@1 with increasing iterations and outperforms a supervised fine-tuned baseline (for the 2B model). LETI is also helpful when the post-processing heuristic is applied to the LM's output: 2B LM improves from 26.89% to 29.53% within two iterations (Tab. 1). We include a qualitative example for the 2B model in Fig. 1.

**Error analysis.** On MBPP test set with 8,000 instances (500 test examples, 16 generations per example), we show how the distribution of error types changes for LETI (2B) in Tab. 1. These error types are concrete exceptions[5] of Python3 programming language. On LETI (2B, w/o post-processing), we initially observed that most errors are SyntaxError (5179, 64.7%) due to no post-processing. We find that LETI can gradually reduce the proportion of generated code that causes SyntaxError by 56.5% (5179 → 652) and produce 63.2% more executable code (pass test + AssertionError). Most of the remaining errors (54.5% out of 71.8%) are due to the generated code being functionally incorrect as validated by the test suite (AssertionError), which can be hard to fix using the error message and stack traces alone (Jones et al., 2002), even for humans. Similarly, on LETI (2B, w/ post-processing), we observe NameError, which can be fixed using the error message alone, is mostly eliminated (810 → 94) within two iterations, demonstrating the effectiveness of LETI. These results also expose the limitation of automated textual feedback from Python interpreter, which can be mitigated by (1) increasing exploration in the hope of finding better code by sampling more per problem (§B.1, Li et al. 2022), (2) leveraging more powerful sources of feedback (Wang et al., 2023b), or (3) keeping pre-training base LM on more relevant solutions.

---

[5]https://docs.python.org/3/library/exceptions.html#concrete-exceptions

Table 1: Count of top-3 error types on MBPP test set before and after LETI fine-tuning.

| LETI (2B) w/o post-processing | | |
|---|---|---|
| | Pre-trained | Fine-tuned |
| # of AssertionError | **1189** | 4356 |
| # of SyntaxError | 5179 | **652** |
| # of IndentationError | 467 | **165** |
| # of Other Errors | 799 | **572** |
| # of Pass Test | 366 | **2255** |
| Pass@1 (%) | 4.50 | **28.00** |
| LETI (2B) w/ post-processing | | |
| | Pre-trained | Fine-tuned |
| # of AssertionError | **3835** | 4376 |
| # of SyntaxError | 437 | **458** |
| # of NameError | 810 | **94** |
| # of Other Errors | **652** | 657 |
| # of Pass Test | 2266 | **2415** |
| Pass@1 (%) | 26.89 | **29.53** |

Table 2: HumanEval performance of LMs finetuned on MBPP using LETI. We observe consistent Pass@10 and Pass@100 improvement across different model sizes. The top-ranked results are presented in **bold**, while the second-ranked results are underlined.

| | HumanEval | | |
|---|---|---|---|
| | Pass@1 | Pass@10 | Pass@100 |
| Pre-trained (350M) | 12.56 | 23.11 | 35.19 |
| LETI (350M) w/o textual feedback | 12.19 | 21.69 | 35.62 |
| LETI (350M) | **13.19** | **23.36** | **36.95** |
| Pre-trained (2B) | **23.70** | 36.64 | 57.01 |
| LETI (2B) w/o textual feedback | 19.90 | 35.62 | 58.48 |
| LETI (2B) | 21.60 | 37.03 | 58.28 |
| LETI (2B, trained w/ post-processing) | 21.60 | **39.51** | **61.46** |

### 3.2.2 HUMANEVAL

**Setup.** We evaluate LM trained on MBPP on another code generation dataset HumanEval (Chen et al., 2021b), which contains 164 handwritten problems to assess language comprehension, reasoning, algorithms, and simple math capabilities. We use the same `pass@k` metric as described in §3.2.1 and apply post-processing for the generated solution.

**Results.** Despite being trained on a problem set MBPP that contains the most basic Python problems, as shown in Tab. 2, LETI can improve LM's capability in other code generation problems in the HumanEval dataset. Compared to pre-trained LM, we observe consistent Pass@10 and Pass@100 improvement across both 350M and 2B LMs, while the 2B LM has a degraded Pass@1 performance. We observe larger improvements for LETI (2B) trained with post-processing as it allows LETI to focus on improving common error (e.g., NameError) in evaluation that applies post-processing.

### 3.3 LEARNING FROM TEXTUAL FEEDBACK IS MORE SAMPLE-EFFICIENT

To study the effect of learning from textual feedback, Fig. 2 compares LETI against a baseline that only uses binary feedback. Regardless of model sizes, LMs trained with textual feedback obtain better final performance and improve faster (up to 2.2x for 2B; Tab. 3).

**LM's ability to leverage textual feedback increases with scale.** A larger model is more effective in learning from textual feedback and can obtain a larger (average) improvement per iteration than a baseline that only uses binary feedback (Tab. 3): 2B model that uses textual feedback improves 2.24x faster than binary feedback, while 350M is only 1.57x faster. Similar to Kaplan et al. (2020), we also find that a larger LM (2B) optimized using LETI obtains larger improvements per iteration (approx. 8x more compared to 350M LM) for both training and testing problems when both are given textual feedback. In other words, a larger model requires fewer gradient updates to achieve similar performance in a smaller model. These observations suggest that we might see more significant gains by applying LETI on LMs of a larger scale (e.g., 6B, 16B), which we leave for future work.

**LMs trained with textual feedback can use samples more efficiently.** As shown in Fig. 3, compared to a baseline that only uses binary feedback, LETI (2B) yields better accuracy and sample efficiency: 2.74x and 2.24x higher improvement rate for $|\mathcal{P}| = 128$ and $|\mathcal{P}| = 374$ (Tab. 4). Interestingly, we observe a different trend for the smaller LM (350M). When decreasing the number of training problems from 374 to 128, LETI actually *underperforms* the baseline that only uses binary feedback. We conjecture that this is because (1) a smaller LM may lack the capacity to learn from textural feedback, and (2) LMs can benefit from a larger $|\mathcal{P}|$ by seeing a more diverse set of problems.

### 3.4 LETI RETAINS REASONING AND CHAIN-OF-THOUGHT PERFORMANCE

**Setup.** We evaluate LETI-optimized LM (w/o post-processing) on additional reasoning tasks, including GSM8K (Grade School Math) Cobbe et al. (2021), a mathematical reasoning dataset that includes grade school math problems, and Big-Bench-Hard (BBH) Suzgun et al. (2022) that includes 26 challenging and diverse tasks (e.g., date understanding, sport understanding) testing

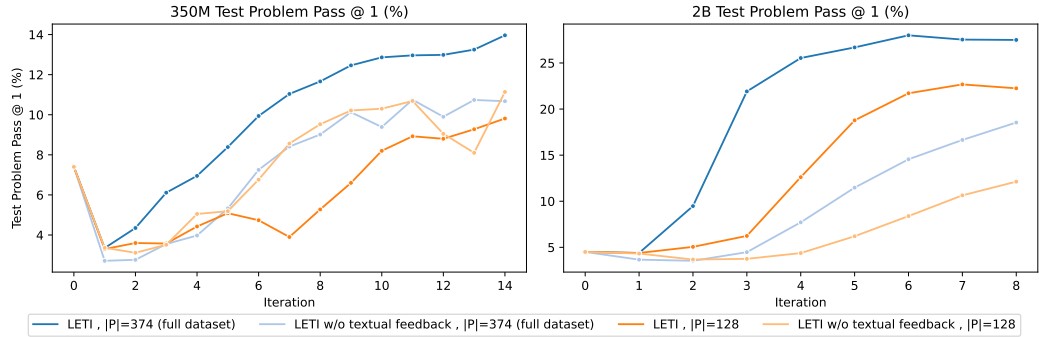

Figure 3: LETI performance with different numbers of training problems $|\mathcal{P}| \in \{128, 374\}$. LETI (2B) with textual feedback can use samples more efficiently than a baseline that does not leverage textual feedback by always achieving higher performance and improvement rate (Tab. 4).

Table 3: On MBPP, LETI improves the LMs' code generation performance by up to 2.24x more per iteration when textual feedback is provided.

| Model Size | Textual Feedback | Initial Pass@1 | Max Pass@1 | #Iter to Max | Avg. improvement per iteration |
|---|---|---|---|---|---|
| 2B | ✓ | 4.50 | **28.00** | 6 | **3.92** (2.24x) |
| | ✗ | 4.50 | 18.54 | 8 | 1.75 |
| 350M | ✓ | 7.40 | **13.96** | 14 | **0.47** (1.57x) |
| | ✗ | 7.40 | 10.75 | 11 | 0.30 |

Table 4: LETI's average improvement per iterations for different numbers of training problems $|\mathcal{P}| \in \{128, 374\}$.

| | | Avg. improvement per iteration | |
|---|---|---|---|
| Model Size | Textual Feedback | # Train Problems $|\mathcal{P}|$ | |
| | | 128 | 374 (full dataset) |
| 2B | ✓ | **2.60** (2.74x) | **3.92** (2.24x) |
| | ✗ | 0.95 | 1.75 |
| 350M | ✓ | 0.17 (0.63x) | **0.47** (1.57x) |
| | ✗ | **0.27** | 0.30 |

model's generic reasoning capability. For GSM8K, we evaluate on `PaL`-style prompting (Gao et al., 2022) settings that ask LM to generate code and execute them to solve the given reasoning problem. Solutions for these reasoning tasks are generated without being conditioned on any reward token (e.g., `<|good|>`). We evaluate Big-Bench-Hard on two prompt settings: direct prompting that asks the model to generate an answer directly and chain-of-thought (CoT) prompting (Wei et al., 2022b) that elicits a series of intermediate reasoning steps from the LM before generating the answer. We calculate the performance gain $\Delta_{\texttt{CoT-direct}}$ from doing chain-of-thought by calculating the performance difference between CoT and direct prompting.

**Results.** As shown in Tab. 5, we observe no significant degradation in out-of-domain reasoning performance (i.e., GSM8K and BBH) after LETI fine-tuning. Moreover, as shown on BBH, applying LETI on a 2B LM improves its chain-of-thought capability compared to its pre-trained checkpoint (i.e., higher `CoT` and $\Delta_{\texttt{CoT-direct}}$). In a smaller 350M model, we observe some degradation in BBH's CoT performance despite also applying regularization via continued pre-training (§2.4).

**Removing regularization degrades performance outside MBPP.** We compare LMs (350M) trained with and without the continued pre-training regularization (§2.4). We observe no significant difference between in-domain task performance (MBPP) shown in Fig. A.9. However, as shown in Tab. 5, removing regularization significantly degrades LM's capability on `PaL`-prompted GSM-8K, similar to findings from Fu et al. (2023), it also degrades BBH's chain-of-thought performance.

Table 5: Performance on additional reasoning tasks, including math reasoning benchmark GSM8K Cobbe et al. (2021) and Big-Bench-Hard (i.e., BBH) Suzgun et al. (2022). *250 out of 6,511 BBH_CoT prompts have more than 2048 tokens, which exceed `CodeGen` models' context window. Scores are set to 0 for these prompts.

| | GSM8K | Big-Bench-Hard | | |
|---|---|---|---|---|
| | PaL | direct | CoT* | $\Delta_{\texttt{CoT-direct}}$ |
| Pre-trained (2B) | 40.03 | 29.67 | **36.81** | 7.14 |
| LETI (2B) | 38.97 | 29.41 | **37.46** | **8.05** |
| LETI (2B, w/ post-processing) | **42.99** | **29.81** | 36.72 | 6.91 |
| LETI (2B) w/o textual feedback | 41.93 | 29.23 | 36.71 | 7.48 |
| LETI (2B) w/o regularization | 32.15 | 30.06 | 35.82 | 5.76 |
| Pre-trained (350M) | 13.04 | **29.10** | **30.53** | **1.43** |
| LETI (350M) | **16.68** | 28.89 | 28.86 | -0.03 |
| LETI (350M) w/o textual feedback | 16.07 | 28.81 | 28.72 | -0.09 |
| LETI (350M) w/o regularization | 7.88 | 28.00 | 28.31 | 0.31 |

### 3.5 LETI IS APPLICABLE TO NLP TASKS LIKE EVENT ARGUMENT EXTRACTION (EAE)

When an NLP task can be formulated into a code generation problem, LETI is equally applicable. We experiment with event argument extraction (EAE), cast as a code generation problem by Wang et al. (2023a). Given an event ontology (Fig. 4 upper left) and a natural language sentence (Fig. 4 bottom left), we ask the LM to generate code to instantiate an event class using correct argument roles extracted from the sentence. Then we can check and examine the instantiated event object to validate the correctness of the solution (Fig. 4, right).

**Solution evaluator implementation.** We build a rule-based solution evaluator for the EAE task that checks the instantiated event object in Python (Fig. 4). Specifically, we first check whether the generation satisfies argument constraints by providing a list of Entity objects for each event argument role (1, 2 in Fig. 4); Then we check whether all the predicted arguments match any of the ground truths (3, Fig. 4) and whether all the correctly identified arguments are classified to the correct event role (4, Fig. 4); Finally, we check if the prediction is complete by identifying all arguments in the ground truth solution (5, Fig. 4). We say the solution is correct with $f_{\text{binary}} = 1$ when the it meets all of the above criteria. Note that the design decision of the solution evaluator (e.g., which error to check first) can influence what type of error LETI-optimized LM will prioritize to avoid.

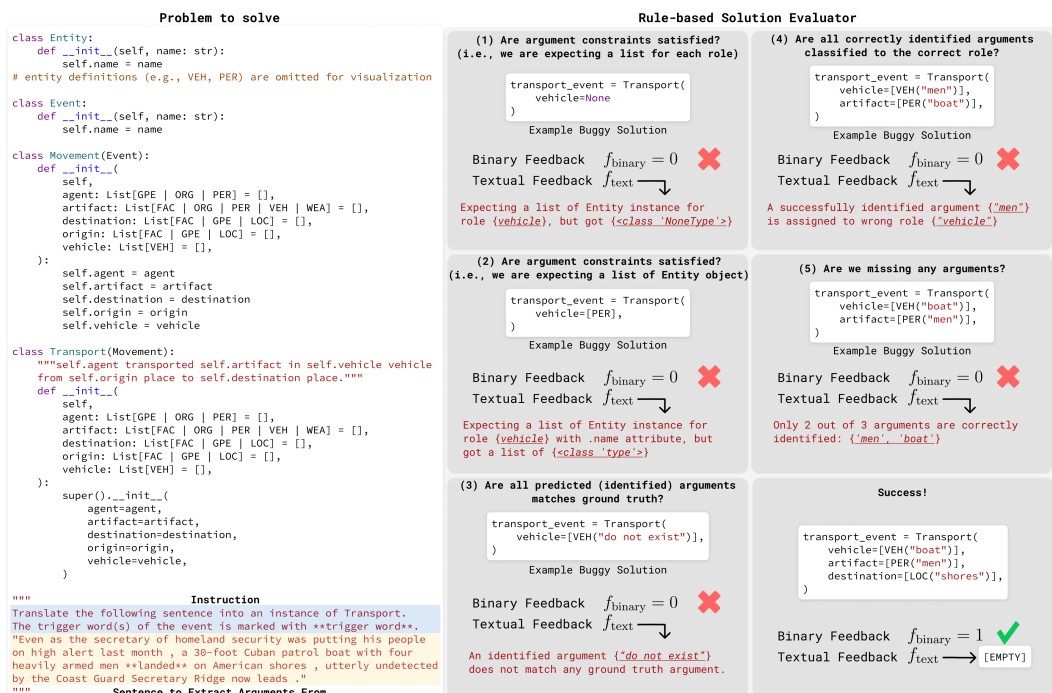

Figure 4: Rule-based Solution Evaluator for Event Argument Extraction (EAE) formulated as code generation task Wang et al. (2023a). Content enclosed by { . . . } in $f_{\text{text}}$ is automatically populated by a Python implementation of Evaluator for any given solution.

**Results.** LETI's performance on EAE task is summarized in Fig. 5. In Fig. 5 (left), We find that LETI is capable of improving the train and test pass rate of generated solutions (i.e., a larger proportion of $f_{\text{binary}} = 1$ for both training and testing test). We also observe increased test performance on task-specific metrics: Argument Identification (Arg-I) F1 increases by $12.3\%$ ($21.2\% \rightarrow 33.5\%$), and Argument Classification (Arg-C) F1 increases $2.6\%$ ($8\% \rightarrow 10.6\%$) with three iterations.

**Implementation of solution verifier could influence the target metric of optimization.** Interestingly, we find that improving $f_{\text{binary}}$ using our solution evaluator results in better performance in some task-specific metrics (e.g., Arg-I and Arg-C precision) but not others (e.g., Arg-I and Arg-C F1). As shown in Fig. 5, Arg-I and Arg-C precision, among other task-specific metrics, has the highest Pearson correlation of 0.93 and 0.73 with test Pass@1, while Arg-I F1 and Arg-C F1 only moderately (0.51) or weakly (0.29) correlate with test Pass@1. One possible reason is that LETI forces the model to be correct on *every* argument it identified in the evaluator implementation (Fig. 4 step 3). This

could inhibit the model from generating arguments very close to the ground truth solutions, reflected in the degrading recall (correlation with Test Pass@1 of -0.08 and -0.24 for Arg-I and Arg-C recall) and improved precision in Fig. 5. This is similar to the reward-shaping problem in reinforcement learning. One can implement solution evaluators that suit better certain metrics.

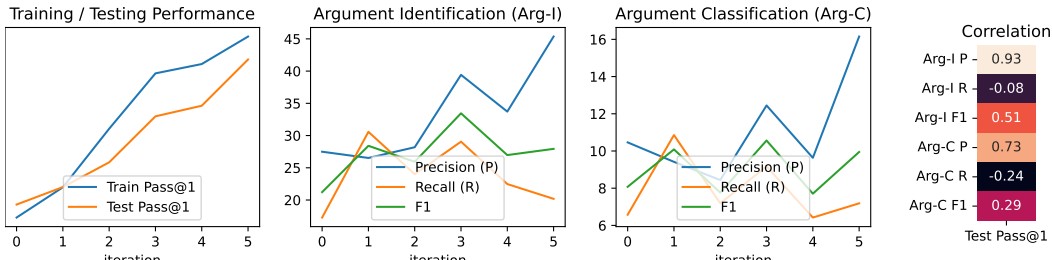

Figure 5: Event Argument Extraction performance and their correlation with Test Pass@1 when using LETI to optimize towards success rate. We found that the rule-based solution evaluator (Fig. 4) can be designed to biased towards optimizing precision as discussed in §3.5.

## 4  RELATED WORK

**Using feedback to improve code generation.** Leveraging non-textual feedback from an interpreter, prior work can generate solutions following natural language instructions by sampling and filtering large amounts of programs (Li et al., 2022; Chen et al., 2022), training a model to rank generated solutions (Inala et al., 2022), fine-tuning a Code-LM on generated solutions verified by test cases (Haluptzok et al., 2022), or training a reward model and using reinforcement learning (RL) to improve Code-LMs (Le et al., 2022). Recent work has explored textual feedback (e.g., error messages, human language feedback) to improve LM for code-related problems. Chen et al. (2023a) improves code generation by fine-tuning the original LM on code refinement generated by conditioning on human language feedback; Different from our work, their fine-tuned LM uses more expensive human feedback and is not trained directly on the provided textual feedback. Chen et al. (2023b); Madaan et al. (2023) improve code generation by allowing LM to look at self-generated (and/or interpreter) feedback; however, the generator LM was frozen and couldn't generate better code on the original problem without these methods, while LETI improves the underlying LM directly.

**Improving LMs with reinforcement learning.** Using PPO, Stiennon et al. (2020); Ouyang et al. (2022) align LMs with human preferences. CodeRL (Le et al., 2022) follows REINFORCE (Williams, 1992) and policy gradient (Sutton et al., 1999) to improve Code-LMs with a scalar reward from the interpreter. Different from LETI that directly leverages textual feedback, these algorithms require either manually crafting (Le et al., 2022) or training (Stiennon et al., 2020; Ouyang et al., 2022) reward/value functions, which could be less scalable for various tasks. Another strand of work leverages Transformer architecture Vaswani et al. (2017) to perform RL with sequence modeling (Janner et al., 2021; Chen et al., 2021a). Lu et al. (2022); Korbak et al. (2023); Zhang et al. (2023); Liu et al. (2023) improve LM by performing condition training, similar to conditioning LM on binary feedback $f_{\text{binary}}$ in LETI. LETI goes beyond the aforementioned work conditioning on the coarse-grained label: we are asking the LM to comprehend and improve directly based on textual feedback (e.g., error messages) that generally contains richer information compared to binary feedback.

## 5  CONCLUSION

We proposed LETI, a new LM fine-tuning paradigm that explores LM's potential to learn from textual interactions. We focused on code generation tasks and showed that one can effectively leverage *automatic* textual feedback from a Python interpreter to improve LMs. Textual feedback outperforms baselines that only use binary feedback in both generation quality and sample efficiency. Furthermore, LETI is equally applicable in NLP tasks that can be formulated as code generation, which we empirically verified on Event Argument Extraction. We refer to §A for a discussion of limitations and future work.

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

# A    LIMITATIONS AND FUTURE WORK

In this study, we only explored the automatic textual feedback from a Python interpreter and did not get the chance to investigate real-world human language feedback which may have higher linguistic diversity and helpfulness. Automatic textual feedback from a Python interpreter can be limited as they are not always useful: as shown in §3.2.1 that they are helpful in improving error types like SyntaxError and NameError. Generally, the stack trace for AssertError (functional correctness) is equivalent to binary feedback telling LM it is wrong but does not provide any additional information. A natural follow-up of LETI would be exploring ways to combine Python interpreter feedback with more helpful feedback (e.g., LLM-simulated feedback Wang et al., 2023b; Madaan et al., 2023), applying to stronger and larger backbone LM (Li et al., 2023; Touvron et al., 2023b), as well as extending to multi-turn setting (Nijkamp et al., 2022).

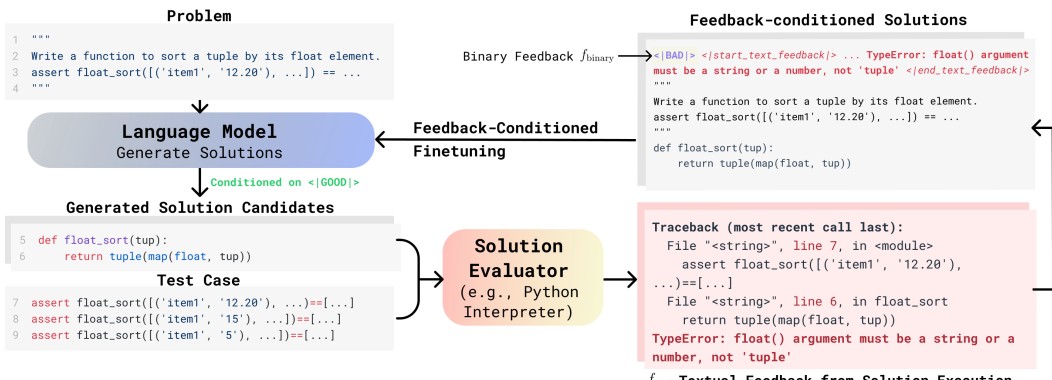

Figure A.6: An LETI Iteration. (1) An actor LM $p_\theta$ generates $n$ solutions for every given problem (§2.1); (2) Each solution $\hat{y}_{i,j}$ for each problem $x_i$ and corresponding test cases $\mathcal{T}_i$ is given to the solution evaluator to obtain binary and textual feedback $F_{i,j}$ on the correctness of $\hat{y}_{i,j}$ on problem $x_i$ (§2.2); (3) The binary and textual feedback $F_{i,j}$ is used to perform feedback-conditioned fine-tuning to improve the actor LM $p_\theta$ (§2.3, equation 1).

# B    ANALYSIS AND ABLATION STUDY

## B.1    DOES THE NUMBER OF SOLUTIONS GENERATED PER PROBLEM MATTER?

We generate different number $n = \{16, 64, 128\}$ of solutions for each given problem. We use $n = 128$ for all other experiments in this paper. In Fig. A.7, we observe that LETI consistently benefits from larger $n$ for each problem (i.e., more exploration).

## B.2    DOES THE NUMBER OF TRAINING PROBLEMS $|\mathcal{P}|$ MATTERS?

In Fig. A.8, we compare an LM trained on a complete MBPP dataset of problems $|\mathcal{P}| = 374$ with LMs trained to iteratively improve on $|\mathcal{P}| = \{16, 64, 128\}$ problems, which corresponds to the first $|\mathcal{P}|$ problems on the MBPP training set.

We observe that the number of training problems impacts the performance of LMs on test sets: larger $|\mathcal{P}|$ generally leads to faster and more significant improvements. LETI can generally improve the 2B model, with a smaller rate of improvement for smaller $|\mathcal{P}|$. However, for the smaller 350M model, we observe net positive improvements on the test set only after the number of training problems exceeds a threshold of $|\mathcal{P}| \geq 128$.

## B.3    HOW DO REWARD TOKENS IMPACT PERFORMANCE?

The LM is fine-tuned on two different reward tokens <|good|> and <|bad|>, which correspond to correct and incorrect solutions (§2.3). In Tab. A.6, we quantify the effect of reward tokens on solution

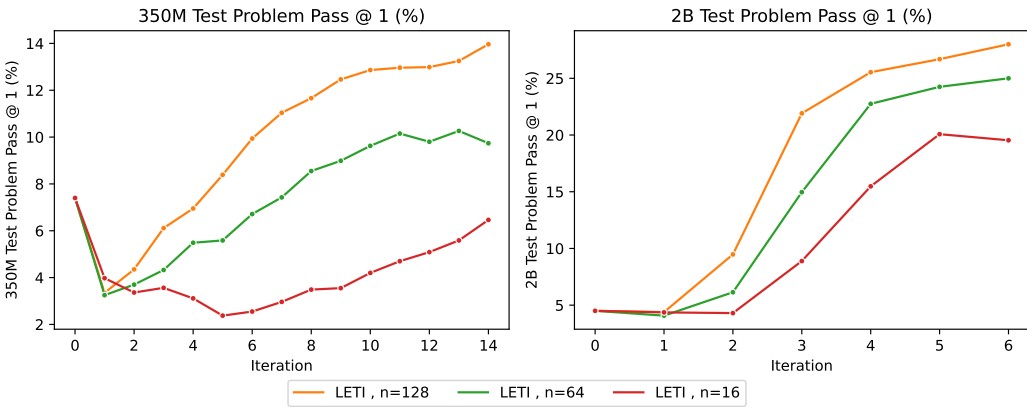

Figure A.7: Comparison of LETI (w/o post-processing) performance when given different numbers $n$ of candidate solutions generated per problem. LETI consistently benefits from larger $n$ for each problem (i.e., more exploration).

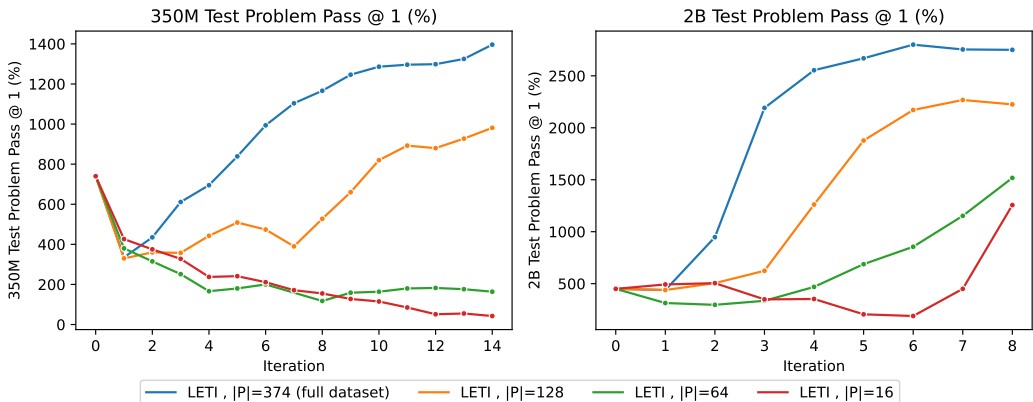

Figure A.8: Comparison of LETI (w/o post-processing) performance when given different numbers $|\mathcal{P}|$ of training problems. Larger $|\mathcal{P}|$ leads to faster and more significant improvements.

quality by calculating the pairwise performance difference between `<|good|>`, `<|bad|>` and `none` (i.e., not conditioned on any reward token). We perform this analysis on two code synthesis datasets MBPP and HumanEval, as well as the math reasoning dataset GSM8K and Big-Bench-Hard, which measures generic reasoning capability.

We find that `<|good|>` generally outperforms `<|bad|>` (i.e., positive $\Delta$`<|good|>` $-$ `<|bad|>`) and both reward tokens outperform `none` on in-domain dataset MBPP. In LETI, the LM is optimized to partition its probability space to put good solutions as sequences that start with `<|good|>` and bad solutions to be sequences starting with `<|bad|>`. This naturally moves solutions that are related to the code synthesis problems away from `none` sequences (i.e., sequences that do not condition on any reward token) towards the space of sequences that start with either `<|good|>` or `<|bad|>`, which could cause the sequences that start with any reward tokens to be better than `none` sequences as we observed.

On the HumanEval code synthesis dataset, we find that conditioning on both reward tokens does not improve performance. Instead, we observe a large gap between `none` and any of the reward tokens, while the performance difference between two reward tokens is minimal. This hints that the solutions for the HumanEval dataset are different compared to in-domain solutions for MBPP, therefore only sequences drawn from the original `none` sequences distribution (i.e., code that an LM has seen during its pre-training) achieves good performance.

We generally observe minimal differences between different reward tokens and `none` on GSM8K and Big-Bench-Hard. That is, performance is similar regardless of whether we are conditioned on any reward token. One notable exception is the `PaL` prompt on GSM8K which performs math reasoning through code generation, where it exhibits a similar pattern of condition on `<|good|>` is better than `<|bad|>` as seen in in-domain dataset MBPP. In fact, somes solutions to GSM8K with `PaL` prompt are very similar to solutions that solve MBPP problems. This suggests that the performance difference between reward tokens could be a way to measure the similarity between two different problems.

Table A.6: Reward Token Analysis. We quantify the effect of reward tokens on solution quality by calculating the pairwise performance difference between `<|good|>`, `<|bad|>` and `none` (i.e., not conditioned on any reward token).

| | MBPP | HumanEval | | | GSM8K | Big-Bench-Hard | |
|---|---|---|---|---|---|---|---|
| | pass@1 | pass@1 | pass@10 | pass@100 | PaL | direct | CoT |
| $\Delta$`<|good|>` $-$ `<|bad|>` | | | | | | | |
| LETI (2B) | 1.00 | -1.11 | -0.39 | -0.13 | 0.91 | 0.05 | -0.14 |
| LETI (350M) | 0.00 | -0.23 | 0.01 | -0.14 | 0.22 | -0.17 | 0.28 |
| $\Delta$`<|good|>` $-$ `none` | | | | | | | |
| LETI (2B) | 16.16 | -17.45 | -29.40 | -48.25 | 1.74 | -0.11 | 0.14 |
| LETI (350M) | 3.54 | -9.85 | -17.24 | -28.44 | -0.61 | 0.02 | 0.09 |
| `<|bad|>` $-$ `none` | | | | | | | |
| LETI (2B) | 15.16 | -16.35 | -29.01 | -48.12 | 0.83 | -0.15 | 0.28 |
| LETI (350M) | 3.54 | -9.62 | -17.25 | -28.31 | -0.83 | 0.18 | -0.18 |

### B.4 DOES THE PERFORMANCE GAIN COME FROM MORE PRE-TRAINING STEPS?

When training LETI, as described in §2.4, we regularize the model by alternating a batch of FCFT (§2.3) with a batch from a continued pre-training batch (§3.1). A natural question arises: Do all the improvements come from FCFT? Is it possible that additional pre-training steps from regularization contribute to the improvements?

We perform an experiment to validate this claim on a 350M model. As shown in Fig. A.10, MBPP test performance cannot improve when only training the LM with more steps of pre-training data; That is, we can attribute LETI's performance improvements to FCFT instead of pre-training regularization.

## C LETI TRAINING DETAILS

For each LETI iteration, we are doing feedback-conditioned fine-tuning for $k = 3$ epochs. We train the 350M model with a learning rate of 1e-5, weight decay of 0.01, and batch size of 128. For the 2B model, we use the same hyperparameter except we change the learning rate to 5e-6 due to instability during training (i.e., spiking loss). Training for 350M and 2B were completed on TPU-v3-8 VM instances. Each iteration (with $k = 3$ epochs) takes approximately 22 hours for 2B model and 4 hours for 350M model.

**Applying LETI to MBPP** Out of 974 total problems in MBPP, it contains 374 training problems, 500 testing problems, and the rest being validation set which we did not use. In every LETI iteration, we generate $n = 128$ solutions for each of the 374 training problems with a sampling temperature of 1.0 to construct our training data for FCFT (§2.3). For test set evaluation, we sample $n = 16$ solutions for each test problem with a sampling temperature of 0.1.

**Applying LETI to Event Argument Extraction (EAE) (§3.5)** We use the ACE-05 dataset following pre-processing as described in Wang et al. (2023a). For each training example, we sample $n = 64$ solutions due to computation capacity limitation. We did not do continued pre-training regularization as described in Fig. 2.4 for more efficient computation since regularization mainly helps maintain out-of-domain performance, which is not the main focus of the EAE experiment.

Figure A.9: Ablation of pre-training data regularization on in-domain task MBPP (§2.4). No significant difference exists in the MBPP test performance for LMs trained with or without pre-training data regularization.

Figure A.10: Ablation of Feedback-conditioned Fine-tuning (FCFT) on in-domain task MBPP (2.3). Doing pre-training data regularization without FCFT does not lead to any improvements.

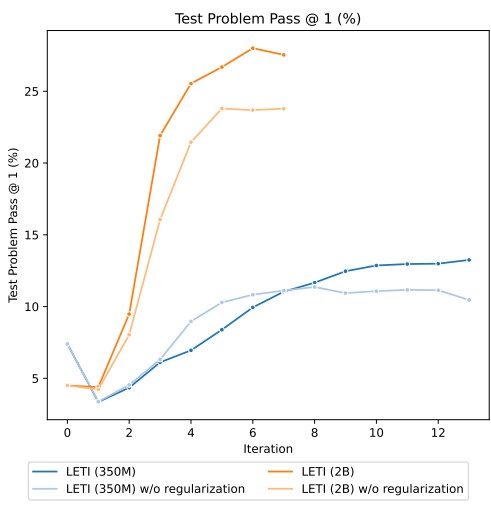

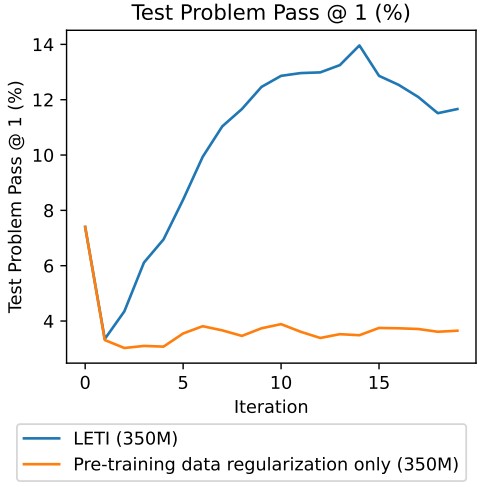

Table A.7: Iteration number of reported LETI-optimized performance in the main paper.

|                                 | $i$-th Iteration |
| ------------------------------- | ---------------- |
| LETI (350M)                     | 14               |
| LETI (2B)                       | 6                |
| LETI (2B, w/ post-processing)   | 3                |

## C.1    METRICS DETAILS

**Pass@k**    We follow the unbiased estimator from Chen et al. (2021b) to estimate `pass@k` that samples $n$ solutions ($n > k$) to more accurately estimate pass@k.

## C.2    EVALUATION DETAILS

We do not condition the generation on any reward token (e.g., `<|good|>`, `<|bad|>`) when generating solutions for the following evaluation datasets.

**GSM-8K**    Following Gao et al. (2022), we use a sampling temperature of 0.7, `top-p` of 0.95, and the number of samples $n = 40$. We generate up to 1,536 tokens for each problem.

**Big-Bench-Hard**    We sample $n = 1$ example for each prompt using a `top-p` of 1 and sampling temperature of 0.0 (deterministic). We generate up to 1,536 tokens for direct prompts and 2,048 tokens for chain-of-thought (CoT) prompts[6]. 250 out of 6,511 CoT prompts have more than 2048 tokens, exceeding the context window of the `CodeGen` models. Scores are set to 0 for these prompts.

**HumanEval**    We follow Nijkamp et al. (2022) to sample $n = 256$ solutions for each problem using `top-p` of 0.95, and temperature of $\{0.2, 0.6, 0.8\}$. The final performance is obtained by taking the max across different temperatures. We generate up to 768 tokens for each problem, which is large enough to include all prompts along with their ground truth solutions.

---

[6]https://github.com/suzgunmirac/BIG-Bench-Hard/tree/main/cot-prompts

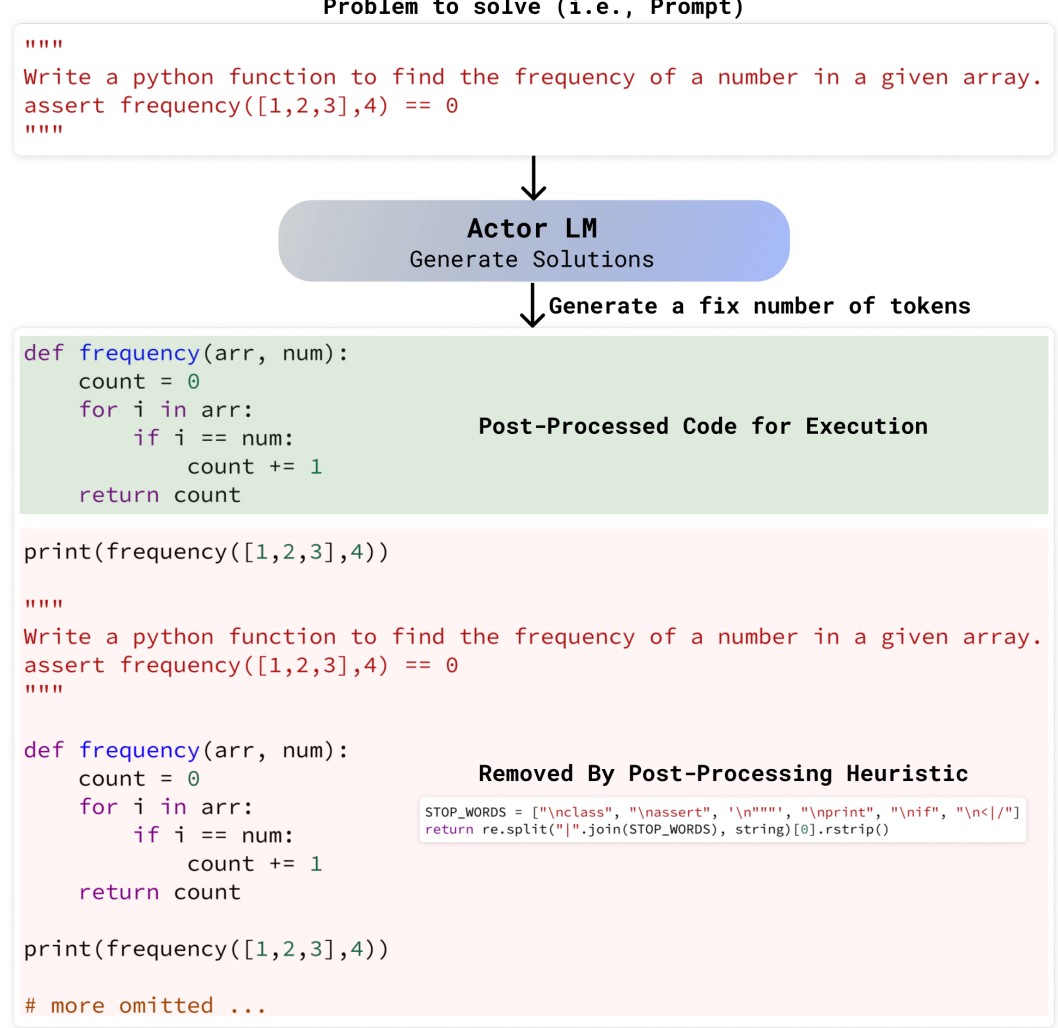

Figure A.11: Examples of code that requires post-processing, generated by pre-trained 2B `CodeGen-mono` on MBPP test set. The LM is asked to generate a fixed number of tokens (up to 512 tokens). It generates a function `frequency`, followed by a print statement. Then it begins to repeat the same prompt and code repeatedly for the rest number of the tokens. Existing implementation typically uses a post-processing heuristic that only keeps the first block of the code (i.e., green block in this figure) for the execution and evaluation. (`https://github.com/bigcode-project/bigcode-evaluation-harness/blob/3ad3b8de11605e74db369450a7ee6704874a4aa7/lm_eval/tasks/mbpp.py#L68`)

## C.3 FINE-TUNED BASELINE DETAILS

**MBPP Fine-tuned Baseline (in Fig. 2)** We fine-tune 350M and 2B `CodeGen-Mono` LM on MBPP training set with 374 examples[7] for 30 epochs with AdamW optimizer of learning rate of 1e-4 and weight decay of 0.01. We evaluate checkpoints (every 6 epochs) on the MBPP test set and report the best pass@1 performance without post-processing. Note that we append `<eos>` token to the end of each ground truth solution for fine-tuning, which encourages the use of `<eos>` to stop the generation when deemed necessary by the LM. The fine-tuned performance is reported in Tab. A.8.

---

[7]`https://huggingface.co/datasets/mbpp`

Table A.8: MBPP Fine-tuned performance. See §C.3 for details.

|  | pass@1 |
|---|---|
| Fine-tuned (`CodeGen-Mono`, 350M) | 16.9 |
| Fine-tuned (`CodeGen-Mono`, 2B) | 20.5 |

