# OpenReview forum: "LETI: Learning to Generate from Textual Interactions"
_ICLR.cc/2024/Conference — ICLR 2024 Conference Withdrawn Submission_

### Official Review · Reviewer_MfcU · 2023-10-30

**Soundness:** 3 good
**Presentation:** 2 fair
**Contribution:** 2 fair
**Rating:** 5
**Confidence:** 3

**Summary:**

The paper introduces LETI, a novel approach to fine-tuning language models using binary and textual feedback -- feedback conditioned finetuning. The work primarily targets code generation tasks. It demonstrates that automatic textual feedback, such as error messages from a Python interpreter, significantly enhances LM performance. Compared to binary feedback only methods, LETI shows superior results in both the quality of generated code and in sample efficiency as evaluated on MBPP and HumanEval dataset. LETI does not require ground truth outputs, only unit tests.

**Strengths:**

LETI addresses a practically relevant problem: The pattern of failure-guided iteration is frequently seen in coding assistants and universal chatbots like ChatGPT, making it a highly relevant use case. LETI's use of detailed textual feedback, such as error messages and stack traces from a Python interpreter, for fine-tuning is a significant strength.

Sample Efficiency in Training: LETI has shown to be more sample efficient compared to traditional fine-tuning methods. By leveraging textual feedback, LETI can achieve better performance with fewer examples and training iterations. This efficiency is particularly advantageous given the computational and data resources required for training LLMs.

**Weaknesses:**

The evaluation process could be improved. A direct comparison of the proposed approach to a PPO-based variant would strengthen the paper. This comparison could provide a clearer understanding of the strengths and limitations of the proposed method in the context of existing advanced techniques.

Limited application to broader NLP tasks: While LETI shows promise in code generation tasks and some NLP tasks framed as code generation problems, such as Event Argument Extraction, its effectiveness in a broader range of NLP tasks remains uncertain. The approach's reliance on feedback akin to that found in programming environments may not translate well to tasks where feedback is less structured or where the notion of "correctness" is more subjective.

**Questions:**

Could you illustrate what the prompts look like for each inference round in the Figure 1? Specifically, does the feedback from iteration $n-1$ get incorporated into the prompt for the subsequent iteration $n$?

Algorithm 1, states that the 'bad' completions are excluded. However, would it be more accurate to state that these 'bad' solutions are only disregarded until they meet the criteria of 'good' on a future iteration? It might be more logical to define 'good' and 'bad' as dynamic terms, changing over iterations.

The observation that exclusively fine-tuning on 'bad' solutions yields subpar outcomes is not unexpected, considering there's no mechanism within the loss function to penalize such solutions.

It would be beneficial to conduct a comparison between the LETI FCFT approach and the PPO-based fine-tuning techniques. Using the Reinforcement Learning from Human Feedback (RLHF) approach with a binary reward system seems to be the most comparable method.

---

> ### Author Response · Authors · 2023-11-19
>
> We thank the reviewer for their responses, and we are glad that the reviewer found our approach highly relevant and sample efficient.
>
> > …Reinforcement Learning based methods are probably state-of-the-art for integrating feedback (whether human-generated or automatic). …it lacks a comparative analysis with RL-based state-of-the-art approaches like RLHF… It would be beneficial to conduct a comparison between the LETI FCFT approach and the RL-based fine-tuning techniques…
>
> The baseline we consider in this paper (LETI w/o textual feedback) *is* a strong online reinforcement learning (RL) algorithm Quark [1], which was shown to perform better than traditional MDP-based RL (e.g., PPO) that requires value function in their paper. LETI directly compares with this stronger baseline Quark and demonstrates the advantage of leveraging textual feedback on both evaluation metrics and sample efficiency (Section 3.3).
>
> [1] Quark: Controllable Text Generation with Reinforced Unlearning
>
>
> > Limited application to broader NLP tasks: While LETI shows promise in code generation tasks … its effectiveness in a broader range of NLP tasks remains uncertain. The approach's reliance on feedback akin to that found in programming environments may not translate well to tasks where feedback is less structured or where the notion of "correctness" is more subjective.
>
> As other reviewers recognized that “such bootstrapping methods are important for further improving LLMs given their data-hungry nature,” and “its successful application to both programming language and natural language tasks suggests that this paradigm can be extended to other domains, making it an impactful contribution to the field of language model fine-tuning,” We argue that deriving an algorithm that can improve code generation and NLP tasks that can be formulated into code generation is already valuable.
>
> As discussed in the introduction, we chose code generation as the main testbed because it can reliably provide automated feedback for any given solution. However, this does not constrain LETI from being applicable to broader NLP tasks with less structured feedback. In fact, we **did not do any post-processing with the textual feedback** (raw error message and stack traces) that comes from the Python Interpreter, and the LM is able to improve upon this textual feedback. Furthermore, in Section 3.4, we show that LETI can work on NLP tasks like EAE by receiving **free-form natural language feedback** (Figure 4, stack trace is NOT provided) as opposed to structured stack traces in code generation tasks. The EAE experiment in Section 3.4 shows that LETI is not constrained by the format of stack traces, and can improve based on textual feedback like free-form natural language.
>
> > Could you illustrate what the prompts look like for each inference round in the Figure 1? Specifically, does the feedback from iteration n−1 get incorporated into the prompt for the subsequent iteration n?
>
> The prompt for each inference round is illustrated as is in Figure 1. That is, the feedback from iteration n-1 will not get incorporated into the prompt for interaction n. Instead, as described in Algorithm 1, the *feedback*, *prompt*, and *solution* from iteration n-1 will be used to fine-tune the LM, and then the fine-tuned LM will generate solutions for iteration n.
>
> > Algorithm 1, states that the 'bad' completions are excluded. However, would it be more accurate to state that these 'bad' solutions are only disregarded until they meet the criteria of 'good' on a future iteration? … The observation that exclusively fine-tuning on 'bad' solutions yields subpar outcomes is not unexpected, considering there's no mechanism within the loss function to penalize such solutions.
>
> Thanks for pointing out this potential confusion! The “bad” completions that did not pass test cases are never thrown away. Instead, all LM’s completions are judged by a solution evaluator (i.e., a test suite) to determine whether they are “good” or “bad” (i.e., binary reward token), then this binary reward token, along with the textual feedback (e.g., error messages and stack traces if the completion is bad) are used to iteratively fine-tune the model (Section 2.2).
>
> We believe the confusion comes from “conditioned on <|good|> for fine-tuned $p_{\theta}$” in the 5th line of Algorithm 1. The reason why we conditioned on a “good” reward token for a LETI-tuned model is that, in previous LETI iterations, we trained the model on both good and bad solutions conditioned on corresponding reward tokens, and our goal is to improve the model to produce more good solutions in the next iteration; therefore, we always sample solutions by conditioning on a good reward token. In the first LETI iteration, the pre-trained model is not trained with LETI; in that case, we just sample the solution from the LM without conditioning on anything. We will make this clearer in the revision.

---

> > ### Comment · Reviewer_MfcU · 2023-11-22
> > **Re: Official Comment by Authors**
> >
> > Thank you for your detailed responses and clarifications. Regarding the comparison to PPO/RLHF: I would appreciate if you could still include a direct comparison of LETI to a PPO based variant, on the same dataset.
> >
> > Overall, your responses have addressed most of my concerns, and I will revise my score accordingly.

---

### Official Review · Reviewer_9ZHv · 2023-11-01

**Soundness:** 3 good
**Presentation:** 4 excellent
**Contribution:** 3 good
**Rating:** 6
**Confidence:** 4

**Summary:**

This paper presents an approach for learning interactively from environment feedback, without accessing the ground truths. Focusing on the task of code generation, the feedback is instantiated as the received error messages from a compiler, or a binary feedback indicating whether or not the generated code passes the given test assertions. The proposed learning algorithm, named LETI, fine-tunes a language model (LM) to generate the task input/output and the feedback. Experimental results showed that LETI can improve the LM to even outperform the supervised fine-tuned baseline when the LM has a larger size, and that a similar idea also applies to the event extraction task (though ground-truth annotations are needed).

**Strengths:**

1. The paper studied an interesting problem of learning interactively from environmental feedback, without needing ground-truth annotations.
2. The experiments are generally solid and comprehensive. When it is applied to a 2B CodeGen LM, LETI improves the LM to even outperform the traditional, human-annotation-required, fine-tuned baseline. This is then supplied by additional analyses confirming the advantage (performance and sample efficiency) of textual feedback compared with using only the binary execution feedback.
3. The paper is well written. Most details are well clarified.

**Weaknesses:**

1. I don't see any significant weaknesses in the proposed approach, but there are a few questions that I would like to have the authors' clarification. See Questions.
2. Missing references. There have been many more works about "using feedback to improve code generation", e.g., the following and their follow-ups or referred papers.
- Elgohary, A., Hosseini, S., & Awadallah, A. H. (2020, July). Speak to your Parser: Interactive Text-to-SQL with Natural Language Feedback. In Proceedings of the 58th Annual Meeting of the Association for Computational Linguistics (pp. 2065-2077).
- Yao, Z., Tang, Y., Yih, W. T., Sun, H., & Su, Y. (2020, November). An Imitation Game for Learning Semantic Parsers from User Interaction. In Proceedings of the 2020 Conference on Empirical Methods in Natural Language Processing (EMNLP) (pp. 6883-6902).

**Questions:**

1. As mentioned in Section 2.1, LETI assumes a code pre-trained LM which can give a decent, initial performance on code generation. I wonder if this is also a reason for the smaller LM benefiting less from LETI (they may be too weak initially)? It can be helpful if the authors could provide the initial good vs. bad instances distribution in each LM's training set.
2. Experiment in Table 5:
- Why PAL for GSM8K but CoT for BBH?
- Could you also show the \delta_{PAL-direct} amount on GSM8K, or is there a reason for not including this result?
- The main experiments say that relatively larger LMs can benefit more from LETI, but in Table 5 LETI shows to hurt the GSM8K performance for 2B (40.03 -> 38.97) while aid for 350M (13.04 -> 16.68). Is this saying that LETI impairs an LM's CoT reasoning performance?
3. The event argument extraction experiment: Can the authors provide results for the supervised, fine-tuned baseline?
As LETI in this application assumes ground-truth annotations, the supervised baseline is necessary for justifying its advantage.

---

> ### Author Response · Authors · 2023-11-19
>
> We thank the reviewer for their detailed and thoughtful responses, and we are glad that the reviewer found our paper interesting, well-written, sound, and easy to follow.
>
> > Missing references. There have been many more works about "using feedback to improve code generation"..
>
> We thank the reviewers for pointing out relevant work! We will include them in the next revision.
>
> > As mentioned in Section 2.1, LETI assumes a code pre-trained LM which can give a decent, initial performance on code generation. I wonder if this is also a reason for the smaller LM benefiting less from LETI (they may be too weak initially)? It can be helpful if the authors could provide the initial good vs. bad instances distribution in each LM's training set.
>
> We think this hypothesis is reasonable; the capability of the base model performance could have an impact on how well each model benefits from LETI. We provide a breakdown analysis of the top-5 most frequent types of LM-generated code (on the training set). We find that the larger model produces more functionally correct instances, which support the hypothesis.
>
> | Type     | 350M   | 2B    |
> |:---------------|:-------|:------|
> | AssertionError | 60.8%  | 47.9% |
> | Success        | 16.2%  | 28.3% |
> | SyntaxError    | 7.0%   | 5.5%  |
> | TypeError      | 6.6%   | 4.0%  |
> | NameError      | 3.1%   | 10.1% |
>
>
> > Experiment in Table 5: Why PAL for GSM8K but CoT for BBH? Could you also show the \delta_{PAL-direct} amount on GSM8K, or is there a reason for not including this result?
>
> The CodeGen model we use has a relatively smaller scale (up to 2B) with limited text-based reasoning capability. Therefore, we follow PaL to reformulate GSM8K as code generation for the model to solve. We did evaluate GSM8K directly without code generation (direct), and there is little distinction before and after LETI fine-tuning as the performance is generally too low. Therefore, we did not include it due to space constraints. We will clarify this in our next revision.
>
> | | GSM8K Direct |
> |:---|:----|
> | 350M Pre-trained |​​ 2.22% |
> | 350M LETI | 2.21% |
> | 2B Pre-trained | 3.34% |
> | 2B LETI | 3.34 % |
>
> > The main experiments say that relatively larger LMs can benefit more from LETI, but in Table 5 LETI shows to hurt the GSM8K performance for 2B (40.03 -> 38.97) while aid for 350M (13.04 -> 16.68). Is this saying that LETI impairs an LM's CoT reasoning performance?
>
> The degradation is not related to CoT, since we use PaL [1] instead of CoT to evaluate GSM8K (i.e., generate code to solve GSM8K question) due to the abovementioned reasons.
> The degradation of the 2B model (w/o post-processing) compared to the base model is minor (40->39) compared to LETI w/o regularization (40->32) on GSM8K.
>
> We believe that the slight performance degradation we observe on GSM8K is potentially related to the slight degradation we observed on the HumanEval of 2B model in Table 2. We hypothesize that training a larger capacity (i.e., number of parameters) model with LETI allows the models to capture more generic patterns of common success and failure solutions reflected in the consistently improved pass@10 and pass@100 performance of HumanEval (generic programming capabilities, similar to recall), while trade-off a bit of pass@1 (e.g., precision) that might cause slight degradation in GSM8K.
>
> [1] PAL: Program-aided Language Models
>
> > The event argument extraction experiment: Can the authors provide results for the supervised, fine-tuned baseline? As LETI in this application assumes ground-truth annotations, the supervised baseline is necessary for justifying its advantage.
>
> We are using ground-truth annotation to simulate automated feedback for EAE. The information the model obtained is less than directly finetuning on those ground-truth annotations since it might take multiple rounds of feedback to guess out the ground-truth annotations.
>
> Therefore, we think LETI is probably a less effective method to improve EAE compared to directly fine-tuning a ground-truth solution (since the construction of the EAE solution evaluator requires ground-truth annotations). Instead of trying to justify the advantages of LETI on EAE, the purpose of Section 3.5 is more to demonstrate the feasibility of LETI to learn from the feedback of different sources (e.g., heuristically constructed natural language feedback) rather than demonstrating the superiority of LETI in the domain of EAE.

---

> > ### Comment · Reviewer_9ZHv · 2023-11-23
> > **Thank you for your response**
> >
> > Thanks to the authors for answering my questions; I agree with other reviewers that the hypothesis about larger models could be better justified with experiments. I've also read a bit about the authors' discussions with other reviewers. The discussion about evaluating LETI w/ post-processing is particularly interesting. The limited improvement of LETI could be an inherent limitation of using the Python interpreter, and in the future this work could be improved by considering multiple sources of feedback. However, the concern about generalization (Reviewer bsvm) seems valid.

---

### Official Review · Reviewer_kgWB · 2023-11-01

**Soundness:** 3 good
**Presentation:** 3 good
**Contribution:** 2 fair
**Rating:** 5
**Confidence:** 4

**Summary:**

LETI is a new language model fine-tuning paradigm that aims to improve LMs' capabilities by learning from nuanced textual interactions with their environment. The authors propose this method for code generation tasks, where the model generates code based on natural language instructions. LETI uses feedback from a Python interpreter to iteratively fine-tune the model on a concatenation of natural language instructions, LM-generated programs, and textual feedback. The authors demonstrate that LETI significantly improves LM's performance on the code generation dataset MBPP without using any ground truth code. Moreover, LETI generalizes to other datasets, such as HumanEval, when trained on MBPP. The authors also find that textual feedback leads to improved generation quality and sample efficiency compared to binary feedback. LETI can be applied to natural language tasks when they can be formulated as code generation problems, which is empirically demonstrated on event argument extraction.

**Strengths:**

1. The paper present comprehensive experiments and evaluations on different datasets and tasks and is well-written and clearly structured.
2. The proposed LETI paradigm holds potential for improving LM's capabilities in various tasks. Its ability to leverage textual feedback for better generation quality and sample efficiency highlights its practical significance, and its successful application to both programming language and natural language tasks suggests that this paradigm can be extended to other domains, making it an impactful contribution to the field of language model fine-tuning.

**Weaknesses:**

1. Evaluation on larger models is needed to provide insights into its scalability and effectiveness.
2. Investigating the impact of different solution evaluator designs on LETI's performance would be informative, as biases may be introduced when optimizing towards certain metrics.
3. Evaluating LETI's effectiveness in other domains and tasks would further validate its generalizability.
4. Comparing LETI with other RL-based approaches that leverage rewards or value functions would help establish the advantages of using textual feedback.

**Questions:**

1. How would LETI perform on larger LM (e.g., 6B or 16B)? Can you provide any insights or expectations on the scalability and effectiveness of LETI when applied to these larger models?
2. Since the performance of LETI relies on the solution evaluator's implementation, could you elaborate on the potential biases that may arise from different evaluator designs? How might these biases impact the performance of LETI in optimizing towards certain metrics?
3. Would it be possible to provide more examples of how LETI can be applied to other domains and tasks? This would help to understand the generalizability of the proposed method across a wider range of applications.

---

> ### Author Response · Authors · 2023-11-19
>
> We thank the reviewer for their detailed responses, and we are glad that the reviewer found our paper useful and easy to follow.
>
> > Evaluation on larger models is needed to provide insights into its scalability and effectiveness… Evaluating LETI's effectiveness in other domains and tasks would further validate its generalizability… How would LETI perform on larger LM (e.g., 6B or 16B)? Can you provide any insights or expectations on the scalability and effectiveness of LETI when applied to these larger models?
>
> We agree that testing LETI on the model of a larger scale (e.g., 6B, 16B) would yield a more comprehensive understanding of the method, and we’d like to do it. However, due to resource limitations, we are not able to run such large-scale experiments. Yet, given the existing possessive results on both in-domain improvements as well as a generalization to HumanEval, we think scaling LETI is likely to bring more benefits than harm. When resource permits, we will include more experiments on other domains.
>
> > Comparing LETI with other RL-based approaches that leverage rewards or value functions would help establish the advantages of using textual feedback…
>
> The baseline we consider in this paper (LETI w/o textual feedback) *is* considered an online reinforcement learning algorithm Quark [1] that performs conditional training on a quantized reward token, which was shown to perform better than traditional MDP-based RL (e.g., PPO) that requires value function in their paper. LETI with textual feedback directly compares with this stronger RL baseline Quark and outperforms it on both evaluation metrics and sample efficiency (Section 3.3).
>
> [1] Quark: Controllable Text Generation with Reinforced Unlearning
>
> > Investigating the impact of different solution evaluator designs on LETI's performance would be informative, as biases may be introduced when optimizing towards certain metrics… Since the performance of LETI relies on the solution evaluator's implementation, could you elaborate on the potential biases that may arise from different evaluator designs? How might these biases impact the performance of LETI in optimizing towards certain metrics? … Would it be possible to provide more examples of how LETI can be applied to other domains and tasks?
>
> We agree that different implementations of a solution evaluator could define different optimization goals. We find the problem of overfitting to a solution evaluator is similar to the problem of reward hacking in reinforcement learning [2]: the policy might overfit to the provided reward function (i.e., a solution evaluator), which might cause the policy to produce a worse outcome on the true reward function we care. In the case of code generation, the test case pass rate we use for LETI training is usually considered the true reward function, and the “reward hacking” phenomenon of LETI is relatively less concerning since it *is* optimizing for a desired goal (i.e., produce functionally correct code).
>
> On domains where the ground-truth reward function (i.e., solution verifier) is challenging to obtain (e.g., need to trade-off between precision and recall for EAE in Section 3.5), different solution verifier designs might impact the optimization goal differently. For example, if we want to improve recall for EAE (as opposed to precision) in Section 3.5, we can instead *not* penalize outputs that hallucinate arguments that do not exist in Figure 4 and count generations that contain all ground-truth arguments as success. Note that LETI is probably a less effective method to improve EAE compared to directly fine-tuning on ground-truth solution (since the construction of the EAE solution evaluator requires ground-truth solution); the purpose of Section 3.5 is to demonstrate the feasibility of LETI to learn from the feedback of different sources (e.g., heuristically constructed) rather than demonstrating the superiority of LETI in the domain of EAE.
>
> A heuristic solution evaluator is less feasible to design for a lot of real-world open-ended NLP tasks (e.g., chit-chat, summarization, information-seeking dialog). Yet, LETI’s framework is still applicable under such settings – LETI can be used to perform Reinforcement Learning from Human Feedback (RLHF, similar to [3]) with the additional benefit of leveraging language feedback: we can employ humans as solution evaluators that provides preference (thumb up or down, i.e., binary reward), and their language feedback as the textual feedback used in LETI. This is similar to our code generation experiment: LETI’s generalizable framework is applied to optimize the true objective/reward function (human preference).
>
> [2] Defining and Characterizing Reward Hacking
>
> [3] Pretraining Language Models with Human Preferences

---

### Official Review · Reviewer_KLdC · 2023-11-08

**Soundness:** 2 fair
**Presentation:** 3 good
**Contribution:** 3 good
**Rating:** 5
**Confidence:** 4

**Summary:**

This paper proposed LETI, a method for improving LLMs by concatenating binary labels and textual feedback with the task I/Os and using it as a dataset for further finetuning. The data collection process for LETI is iterative and automatic, relying only on the test cases and executors (e.g., Python interpreter) and not gold solutions. Experiments on MBPP (in-domain) and HumanEval (out-of-domain transfer) show that incorporating textual feedback is key for improvements yielded by LETI. Extensions beyond code generation tasks are also made to show the generality of the proposed method.

**Strengths:**

S1: The training data can be collected automatically (i.e., no human-in-loop) in an iterative manner without the need for gold-standard solutions, which in principle can easily scale to larger datasets and problem sets. I think such bootstrapping methods are important for further improving LLMs given their data-hungry nature;
S2: The way to construct each training example is quite interesting. Instead of training the models to predict the error message from buggy programs, LETI reverses the order and puts the binary label and textual feedback before the problem instruction and candidate program to learn their association;
S3: A number of ablation studies are presented to show the factors that affect the performance of LETI. I especially like the discussions about the benefits of textual feedback and model sizes;
S4: Though code generation is the main task for LETI, it was also shown to be applicable to non-code tasks.

**Weaknesses:**

W1: The major weakness of this work is the soundness of the experiments. More specifically:
* W1.1: Using *TheStack* as part of the "pretraining dataset". As the authors noted in footnote 4, CodeGen-mono is trained on BigPython (actually it was first trained on the Pile and then BigQuery, then BigPython), and TheStack might contain a substantial amount of code that CodeGen-mono models have never seen before. This could contribute to the performance improvements that are perceived as the effect of "regularization". Note that there are models that are trained on TheStack (e.g., starcoder series), but I do understand that changing the models completely is not feasible. Nevertheless, adding such ablation (i.e., simply further finetune on TheStack w/o any of the proposed methods) in the main results is important.
* W1.2: Comparing results *w/o* post-processing. Note that the post-processing is mostly fixing formatting errors (e.g., semantically wrong solutions won't get corrected just by post-processing), and since $x \bigoplus y$ is a substring of the finetuning data, it is very much expected that it will follow the formatting much better after finetuning. I think comparing the results w/o post-processing will mix the formatting and semantic errors, making it harder to measure the actual improvements from the proposed method. I would suggest showing all the major results *w/ post-processing* (e.g., Figure 2).

W2: One other concern about this work is the cost. If I understand it correctly, for each iteration, LETI needs to perform sampling and finetuning. And from Figure 2, it seems that the majority of the improvements happen after the first 2 iterations.

I hope the authors could clarify these in the discussion period, I am open to adjusting my scores if such concerns are addressed.

**Questions:**

Q1. The models might have already seen such textual feedback (i.e., from Python interpreter) during pretraining, for example from StackOverflow sites, have you tried to apply the iterative method during prompting? Maybe it can be another baseline to be compared with?
Q2. Despite the fact that LETI doesn't need any gold program, can you comment on the computation cost between LETI and simple finetuning (w/ gold program)?

---

> ### Author Response · Authors · 2023-11-19
>
> We thank the reviewer for their detailed responses, and we are glad that the reviewer found our approach important and easy to follow.
>
> > W1.1: Using TheStack as part of the "pretraining dataset". As the authors noted in footnote 4, CodeGen-mono is trained on BigPython… adding such ablation (i.e., simply further finetune on TheStack w/o any of the proposed methods) in the main results is important.‘
>
> We thank the reviewer for the suggestion. In fact, we include the ablation of continued pre-training in Appendix Figure A.10. We find that simply performing continued pre-training for the same amount of training interactions results in **no** performance improvements on MBPP. We will make this clearer in the next revision.
>
> > W1.2: Comparing results w/o post-processing. Note that the post-processing is mostly fixing formatting errors …I think comparing the results w/o post-processing will mix the formatting and semantic errors, making it harder to measure the actual improvements from the proposed method. I would suggest showing all the major results w/ post-processing (e.g., Figure 2).
>
> Thanks for the suggestion! We will include updated figures and results of w/ post-processing in the next revision. We originally did not include it due to space constraints (e.g., the post-processing curve does not fit nicely with the curve w/o post-processing in Figure 2).
>
> On the other hand, despite w/o post-processing potentially mixing the formatting and semantic errors, the LETI still demonstrates consistent advantages over the baseline without textual feedback on both in-domain (i.e., MBPP) and most other metrics (e.g., HumanEval).
>
> > W2: One other concern about this work is the cost. If I understand it correctly, for each iteration, LETI needs to perform sampling and finetuning. And from Figure 2, it seems that the majority of the improvements happen after the first 2 iterations. Q2. Despite the fact that LETI doesn't need any gold program, can you comment on the computation cost between LETI and simple finetuning (w/ gold program)?
>
> This is of similar cost compared to online RL algorithms (e.g., PPO, Policy gradient) that need both sampling & finetuning phases, which are widely applied in RL for improving code generation [1]. In fact, our baseline (LETI w/o textual feedback) is considered a strong online reinforcement learning (RL) algorithm [2] that is trained by conditioned on a reward token.
>
> Similar to how online RL is computationally expensive compared to behavior cloning (e.g., fine-tuning with the gold program), LETI requires at least N times (i.e., the number of generated programs per problem) more computation on fine-tuning per iteration compared to directly fine-tuning on the gold problem, with the additional overhead of sampling N programs.
>
> We expect the cost of LETI could be reduced by designing better sampling algorithms that produce more diverse examples with the same sampling budget (e.g., better exploration); therefore, the model could maintain similar performance improvements with a smaller sampling budget, therefore reducing cost. LETI could potentially be modified into an offline RL setting similar to [3], where we generate a comprehensive set of trajectories that contains various error types and only fine-tune it once to reduce cost.
>
> [1] CodeRL: Mastering Code Generation through Pretrained Models and Deep Reinforcement Learning
>
> [2] Quark: Controllable Text Generation with Reinforced Unlearning
>
> [3] Pretraining Language Models with Human Preferences
>
> > Q1. The models might have already seen such textual feedback (i.e., from Python interpreter) during pretraining…  have you tried to apply the iterative method during prompting? Maybe it can be another baseline to be compared with?
>
> Thanks for the suggestion! Iterative prompting approaches are orthogonal to our approaches since they are applied at **inference time** to iteratively improve a model’s answer to a *particular problem*. However, the fundamental programming capability of the model is not improved (e.g., the next time you ask the model the same question without the context of the error message, the same model would still fail). Instead, LETI aims to improve the model’s fundamental programming capabilities; That is, the model is less likely to make mistakes it has seen before, as demonstrated in Table 1.
> Furthermore, our base model CodeGen is a pre-trained model without going through any alignment (e.g., SFT, RLHF) with limited scale (350M, 2B) and context window (2048), so its ability to engage in a dialog and perform advanced prompting can be limited.
> Therefore, we did not compare with these methods as it is unclear whether we can perform an apple-to-apple comparison on CodeGen. We are happy to hear the reviewers' thoughts on how to perform fair comparisons and include experiments to compare them accordingly.

---

> > ### Comment · Reviewer_KLdC · 2023-11-21
> > **Thanks for the response**
> >
> > I would like to thank the authors for their response.
> >
> > * For W1.1, looking at Figure A.10 actually makes me even more confused. *TheStack* contains much more code data than *ThePile*, which is what CodeGen-NL is trained on. Thus it's quite counter-intuitive that further training it on *TheStack* would actually decrease the performance. Can you explain this part? Maybe I'm missing something here;
> > * For W1.2, thanks for the clarification but I'm not totally convinced and the actual results w/ post-processing will be much more helpful;
> > * For W2 and Q2, the detailed comparison and explanation are great, hope you can add this in the next version of the paper;
> > * Finally for Q1, sorry for the confusion, I was suggesting this baseline as it will be interesting to see the results, I do not believe such a baseline would actually work better than the proposed method as they are only applied at inference time as the authors mentioned.
> >
> > Among these points, the first two are my main concerns, the authors can focus on answering those due to the time limit.

---

> > > ### Author Response · Authors · 2023-11-21
> > >
> > > Thanks for the follow-up!
> > >
> > > > For W1.1, looking at Figure A.10 actually makes me even more confused. TheStack contains much more code data than ThePile, which is what CodeGen-NL is trained on. Thus it's quite counter-intuitive that further training it on TheStack would actually decrease the performance. Can you explain this part? Maybe I'm missing something here;
> > >
> > > Thanks for pointing out this potential confusion! In Section 2.4 and Equation 1, we calculate the loss of one batch of FCFT (1) together with one batch of TheStack pre-training data (2) in LETI. Figure A.10 compares LETI (1+2) with objective (2) only; that is, we only optimize for loss on the pre-training data (2, i.e., continued pre-training) and compare LETI and (2) with *the same number of iterations* to see whether continue pre-training contributes to the performance improvement. That is, **we are not using the entire TheStack** to continue pre-training but only train the LM on the same number of steps as FCFT so that we can make an apple-to-apple comparison.
> > >
> > > We conjecture that continuing pre-training does not improve performance could be due to the fact that the number of gradient update steps (e.g., 374 batches per LETI iteration) we used for both (1+2) and (2) are **much fewer** compared to actual pre-training. For each LETI iteration (x-axis in Figure A.10) with objective (2) only, we train on 2048 (sequence length) * 374 (batches) * 128 (batch size, Appendix C) =98,041,856=98M tokens for continued pre-training. According to StarCoder [1] that trained on TheStack, they use a batch size of 4M tokens, which means each LETI iteration is equivalent to 98M/4M=24.5 train steps in StarCoder pretraining. They pre-train on 1T tokens (1,000,000M/4M=250,000 steps), which is three orders of magnitude larger than ten LETI iterations. We do think if we continue training the LM longer, the general coding capability of the model should increase (evident in the slight upward trend in Figure A.10). We will make this clearer in the next revision.
> > >
> > > [1] StarCoder: may the source be with you!
> > >
> > > > For W1.2, thanks for the clarification but I'm not totally convinced and the actual results w/ post-processing will be much more helpful;
> > >
> > > Here are the results of Figure 2 (w/ post-processing). We find models **w/o textual feedback are less effective** even when purely considering the *semantic errors* (format errors are mostly filtered by post-processing). With a smaller model (350M), a model without textual feedback would not even improve the performance.
> > >
> > > |   iteration |   LeTI (2B, w/ post-processing) |   LeTI (2B, w/ post-processing) w/o textual feedback |   LeTI (350M, w/ post-processing) |   LeTI (350M, w/ post-processing) w/o textual feedback |
> > > |------------:|--------------------------------:|-----------------------------------------------------:|--------------:|-----------------------------------:|
> > > |           0 |                          26.89% |                                               26.90% |        15.29% |                             **15.29%** |
> > > |           1 |                          29.18% |                                          **28.51%** |        14.90% |                             13.56% |
> > > |           2 |                    **29.53%** |                                               28.41% |        15.47% |                             11.91% |
> > > |           3 |                          28.94% |                                               28.29% |        **16.48%** |                             11.94% |
> > >
> > > We conjecture such performance improvement comes from additional information in the textual feedback (e.g., where is the NameError). As discussed in Error Analysis in Section 3.2.1, the limited improvement observed for the 2B model could come from the limitation of the automated textual feedback from the Python interpreter: for harder *semantic errors* like AssertError (i.e., code does not pass the test case), the interpreter is not able to provide additional information beyond a binary label (i.e., fail), which might be less effective compared to more informative error messages for simpler error like NameError (Table 1).
> > >
> > > We include results of LETI (2B, w/ post-processing) in Tables 1, 2, and 5. We are still running evaluations for other models; we will include all these results in the next revision.

---

> > > > ### Comment · Reviewer_KLdC · 2023-12-03
> > > >
> > > > Thanks for answering the follow up questions.
> > > >
> > > > For W1.1, I appreciate the detailed comparison with StarCoder to support the claims that the model might be under-trained in terms of using data from *TheStack*. This could be the case, but also the number of gradient updates across all iterations sums up to 5-7k steps, which should be enough to make a difference in performance. Nevertheless, I find the clarifications helpful and recommend the authors add that in the next version of the paper.
> > > >
> > > > For W1.2, thanks for showing the results for the version w/ post-processing. However, those results seem to suggest that LETI, especially w/ textual feedback, does not help much with the harder errors (e.g., semantically wrong but do not throw any runtime exception), and the improvements doesn't seems signification nor consistent across different model sizes.

---

### Official Review · Reviewer_bsvm · 2023-11-08

**Soundness:** 2 fair
**Presentation:** 4 excellent
**Contribution:** 3 good
**Rating:** 5
**Confidence:** 4

**Summary:**

The paper proposes LETI, which leverages code execution output (stack traces, error messages) in an iterative refinement method to improve a code generation model’s ability to produce correct solutions, and reduce buggy solutions. For a fixed set of epochs, they iteratively finetune the model with a concatenation of code problem prompt, textual feedback (code execution output), and binary feedback (whether execution of generation was successful or not) after an initial code generation, a method they call FCFT (feedback-conditioned finetuning). The proposed method does not require ground truths or instruction-code paired data for finetuning, using just an executor to generate textual feedback. To mitigate distribution shifts due to this finetuning, they interleave the LM objective with the FCFT objective as continued pretraining, doing LM objective optimization on the pretraining data (as regularization), and FCFT on the LETI data.

They conduct FCFT on CodeGen 250M and 2B variants, and trained on MBPP. They demonstrate the usefulness/sample efficiency of text feedback (as opposed to just binary execution feedback) via ablations, and provide results to show how LETI improves generation and sample efficiency over the base CodeGen models for MBPP test set, and also generalization to unseen HumanEval eval set. They test on BIG-Bench-Hard to show that LETI does not regress on base LM performance in general reasoning tasks. Finally, they show that LETI can also be applied to tasks such as event argument extraction and math problem solving, provided the task can be reformulated as a code generation task, and there is an evaluator to provide execution feedback.

**Strengths:**

1) The paper is well motivated, and focuses on how iterative refinement is an important aspect of producing high-quality, correct code solutions. Improving via execution feedback is very interesting and this paper proposes a detailed solution for this.

2) The paper shows relevant gains of LETI on the MBPP test set, over a baseline pretrained model, breaking down the different ways in which it corrects errors such as SyntaxError, NameErrors.

3) The improvement in sample efficiency using textual feedback vs just binary feedback is demonstrated clearly, with a per-iteration average improvement as evidence.

4) The paper provides a good array of evaluations and analysis numbers/plots to demonstrate the performance of LETI, measuring regression (if any) in base performance, generalization performance, breakdown of error types and ablations. Also, the paper has qualitative examples and useful diagrams which makes it clear to understand and follow.

**Weaknesses:**

1) Results for generalization and robustness do not back the claims adequately.

    a. In Table 2, The improvement over HumanEval is mixed for pass@1, with the pretrained model doing better than LETI for 2B. This could be attributed to error (given the small size of HumanEval), so averaging over different runs/seeds would be preferable. Also, if this could be tried on Spider/other code generation tasks this case could be made better.

    b. In Table 5, the pretrained 2B model performs better than LETI 2B without postprocessing, and regression is observed for BigBench-Hard on the pretrained 2B baseline, weakening the stance that base reasoning performance does not regress.

2) LETI hasn’t been tested with larger available CodeGen models (6B, 16B), and given the mixed generalization performance, it becomes difficult to determine whether performance results hold consistently (given that HumanEval results are mixed between 350M and 2B).

3) Results haven’t been compared against strong baselines. For example, methods like [Self-Refine](https://arxiv.org/pdf/2303.17651.pdf) (cited in the paper but not compared), [LEVER](https://openreview.net/pdf?id=Gj3zN9zs4v), and [Self-Debugging](https://arxiv.org/pdf/2304.05128.pdf) are relevant methods which use self-refinement and execution-based training respectively, and it is difficult to gauge how FCFT compares and whether it is beneficial to do it over those methods, given a fixed compute budget.

4) Overall, the main concern is the strength of experiment results. Given that each code generation model needs to be trained via FCFT in the continued pretraining manner, there needs to be more comparison to other recent refinement and execution feedback techniques to illustrate that doing FCFT has a clear performance benefit or is computationally efficient.

**Questions:**

1. In table 2 and table 5, are the pretrained model performances on HumanEval measured with or without postprocessing? I believe a more apples-to-apples comparison would be between Pretrained model w/ postprocessing vs LETI w/o postprocessing. Given that LETI seems to learn how to clean up syntax errors, it seems a more fair comparison.

2. Table 2: Pretrained 2B seems to be better than LETI on pass@1, is there an explanation for why this is so? Results look a bit mixed here. Given the small size of HumanEval, the delta for LETI (350M) is roughly 1 problem. Has this been run over multiple seeds, ensured that this difference is not due to error? Also, given this, would LETI consistently hold over larger model sizes >2B?

3. Although MBPP and HumanEval are different datasets, they focus on similar styles of problems, what about ability to generalize this to other code generation tasks with execution options, like Spider for Text2SQL?

3. This recent paper [LEVER: Learning to Verify Language-to-Code Generation with Execution](https://openreview.net/pdf?id=Gj3zN9zs4v) does a reranking of candidates based on probability that output is correct. A reranker/scoring model is trained for this, and this can be combined with any LM, rather than finetuning a specific LM. Could you comment on the benefits of your method vs this?

4. Any reason as to why “AssertionErrors” and “Other Errors” go slightly up with postprocessing (LETI vs LETI w/postprocessing in Table 1)?

---

> ### Author Response · Authors · 2023-11-19
>
> We thank the reviewer for their detailed responses, and we are glad that the reviewer found our paper well-motivated and easy to follow.
>
> > …a. In Table 2, The improvement over HumanEval is mixed for pass@1... could be attributed to error ... averaging over different runs/seeds would be preferable….
>
> As discussed in Appendix C.1, we follow Codex [1] to sample 256 samples to estimate pass@k using an unbiased estimator to reduce the mentioned randomness. Based on this strategy and observations, we believe the performance trend is consistent across multiple runs, and we will highlight this in the revision.
>
> [1] Evaluating Large Language Models Trained on Code
>
> > b. In Table 5, the pretrained 2B model performs better than LETI 2B without postprocessing, and regression is observed for BigBench-Hard ...
>
> The degradation of the 2B model (w/o post-processing) compared to the base model is minor (40->39) compared to LETI w/o regularization (40->32) on GSM8K and even negligible compared to BBH (less than 0.3%). The general trend established in Table 5 shows that the generalization performance stays the same.
>
> > LETI hasn’t been tested with larger available CodeGen models (6B, 16B) … difficult to determine whether performance results hold consistently…. Table 2: Pretrained 2B seems to be better than LETI on pass@1
>
> We agree that testing LETI on the model of a larger scale (e.g., 6B, 16B) would yield a more comprehensive understanding of the method, and we’d love to do it. However, due to resource limitations, we are not able to run such large-scale experiments. Based on the evidence below, we conjecture that LETI will equally benefit larger models.
>
> As shown in Table 2, regardless of whether post-processing is used during training, we can observe a consistent performance gain for both the 350M model and the 2B model to generalize on Pass@10 and Pass@100 of HumanEval, which we consider to be more reflective of a model’s inherent coding capabilities (e.g., similar to Recall). Such performance gain is even larger when the model is trained with a post-processing heuristic. All these happen without training on any ground-truth solution.
>
> While HumanEval (e.g., Precision) pass@1 did take a minor hit on the 2B model, we hypothesize that training a larger capacity (i.e., number of parameters) model with LETI allows the models to capture more generic patterns of common success and failure programs which sacrificing pass@1 (e.g., precision) for improvement on generic programming capabilities (e.g., recall). Note that such pass@1 (e.g., precision) can be easily improved with established instruction-tuning methods (i.e., alignment) given a base model with good inherent capability, which LETI improves by training the model with both incorrect and correct code solutions with textual feedback.
>
> > Results haven’t been compared against strong baselines. … Self-Refine (cited in the paper but not compared), LEVER, and Self-Debugging are relevant methods which use self-refinement and execution-based training respectively, and it is difficult to gauge how FCFT compares…
>
> These approaches are orthogonal to our approaches since all of these three approaches are applied at **inference time** to improve a model’s answer (e.g., self-refine and self-debug apply iterative prompting, LEVER train verifier to re-rank output programs). However, the fundamental programming capability of the model is not improved (e.g., the next time you ask the model the same question without the context of the error message, the same model would still fail). Instead, LETI aims to improve the model’s fundamental capabilities; That is, the model is less likely to make mistakes it has seen before, as demonstrated in Table 1. Therefore, we did not compare with these methods as it is unclear whether we can perform an apple-to-apple comparison. We are happy to hear the reviewers' thoughts on how to perform fair comparisons and include experiments to compare them accordingly.
>
> > ...could be tried on Spider/other code generation tasks
>
> We had to prioritize the evaluation tasks due to the page limit, and we believe our current tasks properly evaluate the contributions of LETI. According to CodeGen’s original paper [1], its multi-lingual version does not include SQL capability that is required for tasks like Spider. That said, we can include more evaluations based on the CodeGen-Python we used in the next revision when the resource permits.
>
> [1] CodeGen: An Open Large Language Model for Code with Multi-Turn Program Synthesis
>
> > In table 2 and table 5, are the pretrained model performances on HumanEval measured with or without postprocessing?...
>
> HumanEval pre-trained performance is measured **with postprocessing**. We will make this clear in the revision.
>
> > …why “AssertionErrors” and “Other Errors” go slightly up with postprocessing …
>
> The generated program is less likely to have SyntaxError after applying post-processing, therefore could cause other error types to increase.

---

> > ### Comment · Reviewer_bsvm · 2023-11-22
> > **Thank you for the response**
> >
> > Thanks to the authors for their response. Given HumanEval pre-trained performance is measured with postprocessing, it makes the case better. It would be good to have pretrained model performance with and without postprocessing if possible. I'm still concerned about:
> >
> > - "The degradation of the 2B model (w/o post-processing) compared to the base model is minor (40->39) compared to LETI w/o regularization (40->32) on GSM8K and even negligible compared to BBH (less than 0.3%). The general trend established in Table 5 shows that the generalization performance stays the same." : Even though relative regressions between ablations are minor, that still doesn't negate that there is still a regression from the base 2B model. This wouldn't be a major concern had the deltas in Table 2 for Pass@1 and Pass@10 not been that close and mixed themselves.
> >
> > - "While HumanEval (e.g., Precision) pass@1 did take a minor hit on the 2B model, we hypothesize that training a larger capacity (i.e., number of parameters) model with LETI allows the models to capture more generic patterns of common success and failure programs which sacrificing pass@1 (e.g., precision) for improvement on generic programming capabilities (e.g., recall). Note that such pass@1 (e.g., precision) can be easily improved with established instruction-tuning methods (i.e., alignment) given a base model with good inherent capability, which LETI improves by training the model with both incorrect and correct code solutions with textual feedback.": I'm not super convinced by this hypothesis given the experimental results in the paper. To prove improvement in generic programming capabilities it would have been great to see performance on at least 2 evals (HumanEval+ non-MBPP code generation eval), but without an additional eval it becomes difficult to verify this. It would be nice to see this in the next revision.
> >
> > - "These approaches are orthogonal to our approaches since all of these three approaches are applied at inference time to improve a model’s answer...": I agree with your response here that inference time methods may not be the most accurate comparisons. However, It would be good to see a stronger+simpler baseline here to demonstrate the benefits of LETI, such as continued pretraining with theStackv1.1 but without FCFT. This would show that just continued pretraining for a while longer isn't sufficient, and FCFT is crucial. Has any experiment like this been done?
> >
> > Thank you for the rest of the revisions, looking forward to seeing them in the next iteration. My main concerns would be the second 2 points. To sum up my concern: The experimental results still look a bit mixed to me. To have the most fair comparison, it would be nice to see results of:
> > 1. All models without postprocessing.
> > 2. A continued pretraining baseline without FCFT. If this does worse than LETI, I would be convinced of the benefits of FCFT.
> > 3. To back the claim of improved generic code capability over pass@1 tradeoff, it would be great to see at least one other eval (non-MBPP) to justify drop in pass@1 for human eval.
> >
> > Things which are no longer a major concern are comparing with inference time methods, and training on larger size variants.

---

### Official Review · Reviewer_WAJY · 2023-11-12

**Soundness:** 3 good
**Presentation:** 3 good
**Contribution:** 2 fair
**Rating:** 5
**Confidence:** 4

**Summary:**

LeTI finetunes models with additional textual feedback which might be optional - e.g. for code models, failures can generate error messages which can be used as a learning feedback into the model iteratively.

**Strengths:**

Using execution environments' error messages to provide a good/bad feedback split to fine tune is a good strategy and uses a ready resource available as a benefit to code gen models. Such meta data can be further used as future work to assess severity of errors in output which can be used to differentiate multiple outputs from different temperature settings. Any solution which helps bring smaller models up to par with larger models through smarter training is especially beneficial given the resource constraints in inference pathways.

**Weaknesses:**

I would love to see more programming languages handled in this work; it feels very narrowly defined using Python problems and runs the risk of the specific interpreter's error generation capabilities overfitting the solution which might not replicate as we move to other languages or run time environments.

Execution oriented solutions further are heavily dependent on run time environment and specific to it. Otherwise you are limited to syntactic checkers or basic execution approaches which might leave out a lot of errors. Also error messages or stack traces might sometimes be valid in generated code for improper prompts so just their presence is not sufficient to call the generated code as a fail. This is especially true for snippets of code being generated as part of the interactive development scenario which has been most successful for code gen so far.

**Questions:**

- What's the difference between FCFT and RLAF (Reinforcement learning with automated feedback)
- Did you try the solution on other languages than Python? (CodeGen model supports multiple)?
- Did you try on different versions of python interpreter to see if differing error outputs have implications?

**Details Of Ethics Concerns:**

Code models can have significant biases since the pre-training data can to a large proportion focus on few majority viewpoints. Since the improvement here is based on code execution environment's feedback which presumably is based on execution correctness of the program, the work ought not to worsen any such biases.

---

> ### Author Response · Authors · 2023-11-18
>
> We thank the reviewer for their detailed responses, and we are glad that the reviewer found our paper useful and easy to follow.
>
> > Execution-oriented solutions further are heavily dependent on run time environment and specific to it… error messages or stack traces might sometimes be valid in generated code for improper prompts so just their presence is not sufficient to call the generated code as a fail.This is especially true for snippets of code being generated as part of the interactive development scenario which has been most successful for code gen so far.
>
> We thank the reviewer for pointing out the complication of execution-oriented code feedback, especially in the interactive development scenario.
>
> LETI’s goal is to show that it is feasible to allow LM to directly benefit from textual feedback in addition to binary reward (e.g., pass of test cases).
> The current experiment setting (e.g., basic Python programming that is less likely to produce runtime-specific stack traces) we chose is partially limited by the data we have access to: Collecting reliable natural language (i.e., textual) feedback for other domain can be challenging, and controlled Python programming environment allows us to easily obtain automated textual feedback (i.e., stack traces). We are happy to add experiments if the reviewer has any suggestions on how to extend this (e.g., to interactive development scenario)!
>
> > What's the difference between FCFT and RLAF (Reinforcement learning with automated feedback)
>
> We’d appreciate it if the reviewer could provide a specific paper on RLAF.  To the best of our knowledge, as discussed in Section 4, the most closely related line of work that leverage reinforcement learning to improve code generation using code execution feedback (e.g. [1]), which mostly uses **binary/scalar reward** that **requires human-designed heuristics to convert different execution error type** (e.g., SyntaxError, RuntimeError) **into one scalar reward value used to optimize the policy model through reinforcement learning**.
>
> Instead of directly relying on a manually defined reward function that converts traceback errors into scalar reward values, LETI only requires a binary reward (pass test or not) and a textual string of the error message and traceback to perform optimization. This allows us to utilize fine-grained feedback information from stack trace *without a manually designed reward function*, which can potentially scale to different domains that manually designing a reward function for each is not feasible, while informative textual feedback is relatively easy to obtain.
>
> [1] CodeRL: Mastering Code Generation through Pretrained Models and Deep Reinforcement Learning
>
> > I would love to see more programming languages handled in this work; it feels very narrowly defined using Python problems and runs the risk of the specific interpreter's error generation capabilities overfitting the solution… Did you try the solution on other languages than Python? (CodeGen model supports multiple)? Did you try on different versions of python interpreter to see if differing error outputs have implications?
>
> We agree that adding more programming languages would certainly be helpful in providing a more comprehensive evaluation of LETI. Another recent line of work [2] shows that most of today’s LLM can benefit from execution textual feedback (i.e., stack trace and error messages) across different programming languages (Python, C, C++, Go, Java), suggesting that with a good base LLM (i.e., a multi-lingual CodeGen base model), LETI can scale to different languages. We plan to add evaluation across different languages and interpreter versions when resources permit.
>
>
> [2] Understanding the Effectiveness of Large Language Models in Code Translation

---

> > ### Comment · Reviewer_WAJY · 2023-11-23
> > **Thank you authors for your response**
> >
> > Thanks for your detailed responses. I continue to worry about the stack trace effectiveness but since pass/fail is primary, it might work. I have updated my review scores accordingly. Please keep up the good work. As more languages get added, the appeal will get broad.